# Coupling aerosols to (cirrus) clouds in the global aerosol-climate model EMAC-MADE3

Mattia Righi[1], Johannes Hendricks[1], Ulrike Lohmann[2], Christof Gerhard Beer[1], Valerian Hahn[1], Bernd Heinold[3], Romy Heller[1], Martina Krämer[4,5], Michael Ponater[1], Christian Rolf[4], Ina Tegen[3], and Christiane Voigt[1]

[1]Deutsches Zentrum für Luft- und Raumfahrt (DLR), Institut für Physik der Atmosphäre, Oberpfaffenhofen, Germany
[2]Institute for Atmospheric and Climate Science, ETH Zürich, Zürich, Switzerland
[3]Leibniz Institute for Tropospheric Research (TROPOS), Leipzig, Germany
[4]Research Centre Jülich, Institute for Energy and Climate Research 7: Stratosphere (IEK-7), Jülich, Germany
[5]Johannes Gutenberg-Universität, Institut für Physik der Atmosphäre, Mainz, Germany

*Correspondence to:* Mattia Righi (mattia.righi@dlr.de)

**Abstract.** A new cloud microphysical scheme including a detailed parameterization for aerosol-driven ice formation in cirrus clouds is implemented in the global chemistry-climate model EMAC and coupled to the aerosol submodel MADE3. The new scheme is able to consistently simulate three regimes of stratiform clouds – liquid, mixed- and ice-phase (cirrus) clouds – considering the activation of aerosol particles to form cloud droplets and the nucleation of ice crystals. In the cirrus regime, it allows for the competition between homogeneous and heterogeneous freezing for the available supersaturated water vapor, taking into account different types of ice-nucleating particles, whose specific ice-nucleating properties can be flexibly varied in the model setup. The new model configuration is tuned to find the optimal set of parameters that minimises the model deviations with respect to observations. A detailed evaluation is also performed comparing the model results for standard cloud and radiation variables with a comprehensive set of observations from satellite retrievals and in situ measurements. The performance of EMAC-MADE3 in this new coupled configuration is in line with similar global coupled models and with other global aerosol models featuring ice cloud parameterizations. Some remaining discrepancies, namely a high positive bias in liquid water path in the northern hemisphere and overestimated (underestimated) cloud droplet number concentrations over the tropical oceans (in the extra-tropical regions), which are both a common problem of this kind of models, need to be taken into account in future applications of the model. To further demonstrate the readiness of the new model system for application studies, an estimate of the anthropogenic aerosol effective radiative forcing (ERF) is provided, showing that EMAC-MADE3 simulates a relatively strong aerosol-induced cooling, but within the range reported in the IPCC assessments.

## 1 Introduction

The impact of aerosol on atmospheric composition and climate still represents one of the largest uncertainties in the quantification of anthropogenic climate change (Boucher et al., 2013). Aerosol particles influence the Earth's radiation budget via scattering and absorption of incoming solar radiation (aerosol-radiation interactions) or indirectly by changing cloud microphysical and radiative properties (aerosol-cloud interactions). The level of scientific understanding of the underlying processes

is still relatively low and their representation in global models, which are the only available tools for estimating the respective climate impacts, is challenging.

This is particularly the case for the investigation of aerosol-cloud interactions, which requires a detailed knowledge of various processes acting on a wide range of spatial and temporal scales. Aerosol particles can act as cloud condensation nuclei

(CCN) for the formation of cloud droplets in liquid clouds (e.g., Andreae et al., 2005; McFiggans et al., 2006). This process is controlled by the microphysical properties of the CCN (such as number concentration, size and chemical composition), but also depends on the mesoscale and large-scale atmospheric dynamics, which determine the occurrence and strength of vertical updrafts, leading in turn to cooling of the rising air parcels and supersaturation of water vapor available for condensation. In recent years, significant progress has been made in developing parameterizations for describing the aerosol activation process

in liquid clouds in the framework of global models (see Ghan et al., 2011, for a review), but the uncertainties remain large.

Even more complex is the aerosol-induced formation of ice crystals. At atmospheric conditions, the direct freezing of supercooled liquid solutions requires a very high relative humidity, above $\text{RH}_{\text{ice}} = 140\%$ (i.e., a saturation ratio over ice greater than 1.4). This process is called homogeneous freezing and can only occur at temperatures below the so-called homogeneous freezing threshold, around 235 K (Koop et al., 2000). At lower supersaturations (or higher temperatures), ice crystals form

in the presence of ice-nucleating particles (INPs), which reduces the energy barrier for initiating the freezing process, thus lowering the relative humidity threshold for the formation of ice crystals. This process is collectively termed heterogeneous freezing, but it actually occurs along several formation pathways, depending on the properties of the involved INP and on the supersaturation (Vali et al., 2015): immersion freezing (initiated by an INP immersed in a cloud or solution droplet), contact freezing (initiated by the collision of a supercooled water droplet with a solid INP), condensation freezing (condensation of

water vapor on the surface of an INP and subsequent, almost concurrent, freezing), and deposition nucleation (direct deposition of water vapor on the surface of an INP). Recent studies also discussed pore condensation and freezing on porous INPs as a further ice formation pathway (Wagner et al., 2016; Marcolli, 2017; Mahrt et al., 2018; David et al., 2019). Only a small subset (0.01-0.001%) of the aerosol particles in the atmosphere can act as INPs. In situ measurements and laboratory studies showed that in particular mineral dust, black carbon, organic and biogenic particles can serve as INPs across a wide range of

ice supersaturations (Hoose and Möhler, 2012; Cziczo et al., 2013; Kanji et al., 2017), but large uncertainties still exist on the freezing properties of the aerosol particles. The role of INPs is particularly complex in the cirrus regime, i.e. at temperatures below the homogeneous freezing threshold, since they lead to a competition between homogeneous and heterogeneous freezing for the available supersaturated water vapor. The dominance of either process over the other is essential in determining the number concentration and size of the formed ice crystals, affecting the properties of the cirrus clouds (Kärcher et al., 2006;

Spichtinger and Gierens, 2009).

In this paper, we describe and evaluate the implementation of a new two-moment cloud microphysical scheme (Kuebbeler et al., 2014) into the global model EMAC (ECHAM/MESSy Atmospheric Chemistry; Jöckel et al., 2010) and its coupling to the aerosol microphysics submodel MADE3 (Modal Aerosol Dynamics model for Europe adapted for global applications, third generation; Kaiser et al., 2019). The new cloud scheme is based on a previous scheme by Lohmann and Hoose (2009) already

available in EMAC, but it now includes the parameterization of aerosol-induced cirrus cloud formation by Kärcher et al. (2006).

This parameterization describes the ice formation processes in cirrus clouds depending on the properties of the INPs, while accounting for the competition between homogeneous and heterogeneous ice formation. Whereas Kuebbeler et al. (2014) only considered heterogeneous freezing of mineral dust, in this study we further extend the parameterization to also account for black carbon (BC) as a possible INP.

5 The ice-nucleating properties of the different types of BC are still highly debated, but several laboratory studies suggest that BC may act as INP at typical cirrus temperatures. Möhler et al. (2005a, b) investigated the ice nucleating properties of coated and uncoated soot in the AIDA cloud chamber and found that uncoated soot is able to nucleate ice at low ice saturation ratios between 1.1 and 1.3, but pointed out that coating with sulfuric acid and mixing with organic carbon increase the ice supersaturation threshold. This is supported by the measurements of Crawford et al. (2011), who also used the AIDA cloud 10 chamber to study the ice nucleation of coated and uncoated propane flame soot and reported a 1% nucleated fraction of uncoated low organic carbon soot at saturation ratios as low as 1.22, while they measured lower ice formation efficiencies for soot with higher organic carbon content and for coated soot. Koehler et al. (2009) analysed different soot types and observed ice nucleation below the homogeneous freezing threshold for some of them (including TC1 soot resulting from burning of aviation kerosene). In an intercomparison study among different instruments, Kanji et al. (2011) reported ice nucleation on 15 graphite soot at supersaturations between 1.3 and 1.5. Chou et al. (2013) considered fresh and aged diesel soot particles and measured ice nucleation fractions of several percent at ice saturation ratios around 1.4. Also Kulkarni et al. (2016) analysed the ice formation ability of diesel soot under cirrus conditions and reported a 1% frozen fraction of the soot particles at similar ice saturation ratios. Nichman et al. (2019) examined six types of BC particles considered as proxies for atmospheric BC and found onset saturation thresholds for ice nucleation between 1.1 and 1.5. Recent studies observed BC nucleation at cirrus 20 temperatures, but explained it with pore condensation and freezing rather than with deposition nucleation (Wagner et al., 2016; Marcolli, 2017; Mahrt et al., 2018; David et al., 2019). As shown by Mahrt et al. (2019), the process of pore condensation and freezing will become more important after cloud processing of soot, which enhances soot-induced heterogeneous ice formation at cirrus temperatures by reducing the saturation threshold for ice nucleation.

Despite the uncertainties resulting from the large range of measured ice formation abilities of BC in the cirrus regime, the 25 laboratory studies clearly reveal that the effects of soot on ice cloud formation are potentially relevant and the resulting climate impacts could be significant, especially when considering specific emission sources such as aviation (Koehler et al., 2009; Hendricks et al., 2011; Zhou and Penner, 2014; Penner et al., 2018; Urbanek et al., 2018) or land transport (Chou et al., 2013; Kulkarni et al., 2016).

There are, however, only few global models capable of simulating aerosol-induced ice formation in the cirrus regime in 30 detail. Liu and Penner (2005) developed a parameterization for homogeneous and heterogeneous ice nucleation, which was later implemented in the CAM model by Gettelman et al. (2010). More recently, Bacer et al. (2018) coupled the Global Modal-aerosol eXtension (GMXe) submodel (Pringle et al., 2010) to the cirrus parameterization by Barahona and Nenes (2009) in the EMAC model, opening interesting perspectives for comparing different aerosol and cloud microphysical schemes within the same model framework. Several studies attempted to quantify the climate impact resulting from the influence of BC on cirrus 35 clouds, but these estimates are quite diverse and there is no consensus on the magnitude, and not even on the sign, of this

effect. Using the CAM3 model coupled with the IMPACT aerosol model, Liu et al. (2009) simulated the impact of soot on cirrus clouds and found a significant warming effect, strongly dependent on the assumptions on the ice nucleating ability of soot itself. Using offline calculations, Penner et al. (2009), however, argued that the soot impact on cirrus clouds results in a significant cooling effect, while Hendricks et al. (2011, with the ECHAM4 model) and Gettelman and Chen (2013, with the CAM5 model) found no statistically significant climate effects. Zhou and Penner (2014) discussed the role of background INPs as an important source of uncertainty affecting the estimates of aviation soot impacts on climate. Penner et al. (2018) included an ice-formation parameterization for cirrus clouds in the NCAR-CAM5.3 model coupled to the IMPACT aerosol model, also distinguishing three aerosol mixing states in three size modes. Their resulting estimates of the radiative forcing from various emission sources show a large negative climate impact due to aerosol-induced cirrus modifications.

The compelling need for additional insights into this issue motivated the extension of MADE3 towards a better resolved representation of INP properties (Kaiser et al., 2019) and the coupling to a new cloud scheme with a detailed parameterization for aerosol-induced ice formation in cirrus clouds, which is described in the present paper. Since, as discussed above, experimental support for the ice-nucleating properties of BC is still very limited, the modeling tools need to be designed in such a way that different assumptions can be flexibly and efficiently assessed by means of sensitivity studies, in order to explore the parameter space and provide a more precise uncertainty estimate for the resulting effects on climate.

The new model configuration is tuned to obtain an optimal agreement in the representation of key cloud and radiation variables in comparison with observations. The tuned setup is then evaluated in detail against a wide range of satellite, ground-based and aircraft data. As a first example of application we simulated the anthropogenic aerosol ERF effect from anthropogenic emissions with respect to pre-industrial times. This estimate will serve as a basis for future application studies on specific sectors, for which the new model configuration described here is specifically designed. One of the main application targets for this model will be the improvement of the current estimates on the climate impact of the transport sectors (Righi et al., 2011, 2013, 2015b, 2016), also considering the role of cirrus clouds, which motivates the consideration of BC as possible INP in the cirrus parameterization implemented here.

The paper is organized as follows: the EMAC-MADE3 model and its configuration is described in Sect. 2. The implementation of the new cloud scheme with the cirrus parameterization is detailed in Sect. 3. Section 4 discusses the model tuning and the comparison with observations. For demonstration, an application of the new model configuration is briefly presented in Sect. 5, where the simulated anthropogenic aerosol ERF is calculated and compared with the IPCC estimates. A summary of the main conclusions of this work is then given in Sect. 6.

## 2   Model description and configuration

We use the EMAC global model with the aerosol submodel MADE3 in the same setup as described in Kaiser et al. (2019, hereafter K19), but with an explicit representation of the interactions of aerosols with clouds and radiation, which is crucial for the present paper and for the planned follow-up studies. In this section, we briefly summarize the main features of EMAC-MADE3 and discuss only the main differences with respect to the uncoupled model configuration of K19.

EMAC is a numerical chemistry and climate simulation system that includes submodels describing tropospheric and middle atmospheric processes and their interaction with oceans, land and human influences (Jöckel et al., 2010). EMAC is based on the second version of the Modular Earth Submodel System (MESSy) to link multi-institutional computer codes. The core atmospheric model is the ECHAM5 (5th generation European Centre Hamburg) general circulation model (Roeckner et al., 2006). For the present study we apply EMAC (ECHAM5 version 5.3.02, MESSy version 2.54) in the T42L41 resolution, i.e., with a spherical truncation of T42 (corresponding to a quadratic Gaussian grid of $\sim 2.8°$ by $2.8°$ in latitude and longitude) and with 41 vertical hybrid $\sigma$-pressure levels up to 5 hPa. This resolution has been successfully used in previous studies with a focus on processes in the upper troposphere / lower stratosphere (Dietmüller et al., 2014; Bock and Burkhardt, 2016). The model time-step length $\Delta t$ for this resolution is 15 minutes. Unless otherwise specified, the model output is stored with a temporal resolution of 11 hours, which on average samples the full daily cycle.

Aerosols are simulated using the aerosol submodel MADE3 (Kaiser et al., 2014), considering aerosol sulfate, ammonium, nitrate, sodium, chloride, particulate organic matter, black carbon, mineral dust, and aerosol water. These compounds are assumed to be distributed over 9 modes, covering three size classes (Aitken, accumulation and coarse) and three particle mixing states, namely: soluble particles, insoluble particles (i.e., particles mainly composed of insoluble components, such as mineral dust or soot) and mixed particles (soluble compounds with insoluble immersions). This detailed description of particle mixing allows an advanced representation of aerosol-induced ice formation in the troposphere via different processes, which is the main focus of the current study.

As mentioned above, we apply here the same model configuration as K19, except for the submodels controlling the coupling between aerosol, clouds and radiation in the model, which are now configured to enable such coupling. The CLOUD submodel, which deals with cloud microphysics and precipitation formation in stratiform clouds at all levels, including aerosol effects on warm and mixed-phase clouds, uses the two-moment cloud scheme by Kuebbeler et al. (2014, hereafter K14) instead of the standard ECHAM5 scheme by Roeckner et al. (2006), and is described in detail in Sect. 3. Cloud-radiation and aerosol-radiation interactions are now explicitly simulated providing the corresponding coupling parameters to the submodels CLOUDOPT (cloud cover, cloud liquid and ice water content, cloud droplet and ice crystal effective radii) and RAD (aerosol optical thickness, asymmetry factor, and single scattering albedo in the respective model layers, as calculated by the AEROPT submodel). Since the present model configuration is designed to study the radiative effects of aerosol and clouds, the concentrations of radiatively active gases other than water vapor (i.e., $CO_2$, $CH_4$, $N_2O$, $O_3$ and chlorofluorocarbons) are prescribed by means of globally constant distributions in RAD. Further details on these submodels as part of the radiation scheme of EMAC are provided in Dietmüller et al. (2016).

The reference simulation evaluated in this work covers a time period of 10 years, from 1996 to 2005, with an additional year (1995) as spin-up. The tuning experiments discussed in Sect. 4.1 are limited to 3 years (1999–2001, with 1998 as spin-up) to reduce the computational costs. As in K19, model dynamics is nudged by relaxing wind divergence and vorticity, temperature, and the logarithm of the surface pressure towards the ERA-Interim reanalysis data (Dee et al., 2011) along the simulated time period. Anthropogenic and biomass burning emissions of both gases and aerosols are prescribed according to the CMIP5 inventory of Lamarque et al. (2010) for the year 2000. Volcanic emissions of sulphur dioxide and primary aerosol sulfate are

taken from the AeroCom inventory (Dentener et al., 2006). Wind-driven sea-salt emissions are calculated online according to the parameterization by Guelle et al. (2001). Further details about the emission setup are given in Sect. 2.4 of K19.

In contrast to K19, where dust emissions were prescribed via an offline climatology, namely the AeroCom dust climatology for the year 2000 (Dentener et al., 2006), we now apply the online parameterization developed by Tegen et al. (2002). This parameterization calculates dust emissions from 192 internal dust size classes ranging from 0.2 to 1300 $\mu$m according to the prognostic 10-meter-wind-speed and prescribed external input fields for dust source areas, soil types and vegetation cover (see Tegen et al., 2002; Stier et al., 2005; Cheng et al., 2008; Gläser et al., 2012, for more details). Mass emission fluxes of the single size classes are then grouped in two modes, which we assign to the MADE3 insoluble accumulation and coarse modes. The corresponding number emissions are then derived assuming a log-normal size distribution, with median diameters 0.42 and 1.30 $\mu$m, and mode widths sigma of 1.59 and 2.0 geometric standard deviations, for the accumulation and the coarse mode, respectively, following Dentener et al. (2006). In order to obtain a reliable representation of dust emissions with the T42 resolution used in this work, the model has been re-tuned with respect to Gläser et al. (2012) by adjusting the wind stress threshold for dust emissions as described in Tegen et al. (2004). A value of 0.688 is chosen for this parameter, in order to match the total dust emission in the AeroCom inventory for the year 2000, i.e. $\sim 1760$ Tg yr$^{-1}$, of which about 1.2% (98.8%) are emitted in the accumulation (coarse) mode. We use this dataset as a reference since it is well evaluated and widely used in several global modeling studies (see Huneeus et al., 2011). An additional correction is introduced to avoid artefacts of extremely high emissions in model grid boxes near the Himalaya region. These local artefacts dominate global dust emissions and are up to 1000 times higher in the current setup than the corresponding values in the AeroCom dataset. Due to the relatively low spatial model resolution, strong dust sources in this region (namely the Taklamakan desert) coincide with high surface winds (resulting from the steep orography gradient at the northern slope of the Himalayas) within the same model grid box. This results in unrealistically high dust emissions in such grid boxes, as also noted by Gläser et al. (2012), who found that these artefacts vanish at horizontal resolutions of T85 and higher. In our setup, these artifacts are removed by setting a threshold height for the orography, above which emission fluxes are set to zero. This is set to 4000 m for the present resolution.

## 3   Cloud microphysical scheme and coupling to aerosol

In the present study we use a detailed cloud microphysical scheme which describes aerosol-driven formation of cloud droplets and ice crystals. An important feature of this cloud scheme is a detailed parameterization of aerosol-induced formation of ice crystals in the cirrus regime (Kärcher et al., 2006; Hendricks et al., 2011). The cloud scheme was originally developed by K14 for the ECHAM5 model and coupled to the Hamburg Aerosol Model (HAM, Stier et al., 2005). Here we implement the cloud scheme in the MESSy framework and couple it to MADE3. With respect to HAM, MADE3 provides a more detailed description of aerosol mixing states, using 9 instead of 7 modes explicitly distinguishing between purely soluble, purely insoluble and mixed particles. This is especially important for the ice phase, since formation of ice crystals in the troposphere can occur along different pathways, depending on the properties of the INPs that initiate the process.

The cloud scheme solves prognostic equations for cloud liquid water content, ice water content, cloud droplet and ice crystal number concentrations, considering aerosol-induced formation of cloud droplets and ice crystals, as well as rain and snow formation, condensational and depositional growth, evaporation of cloud water and rain, sublimation and melting of cloud ice and snow, freezing of cloud water, as well as sedimentation of cloud ice (Lohmann et al., 2007, 2008). Cloud cover is treated diagnostically using the Sundqvist scheme, which assumes partial cloud cover above a critical threshold of relative humidity and full coverage at saturation (Sundqvist et al., 1989). Sub-grid scale variability of the vertical velocity, i.e. vertical updrafts, which cannot be resolved by the global model due to its coarse resolution, is accounted for as in Lohmann and Kärcher (2002) by adding a turbulent component $\omega_t$ to the large-scale vertical velocity $\omega_{ls}$ proportional to square root of the turbulent kinetic energy (TKE):

$$\omega = \omega_{ls} + \omega_t + \omega_{gw} = \omega_{ls} + c\sqrt{\text{TKE}} + \omega_{gw} \tag{1}$$

Here, we choose $c = 1.33$ for liquid and mixed-phase clouds (Lohmann et al., 2007) and $c = 0.7$ for cirrus (Kärcher and Lohmann, 2002). As in K14, we also consider the effect of orographic gravity waves on the vertical velocity, by adding a further term $\omega_{gw}$ to the right hand side of Eq. (1). In EMAC, this component is calculated by the submodel OROGW which implements the parameterization by Joos et al. (2008) originally developed for ECHAM5 and used in K14.

In the cloud scheme three different regimes of stratiform clouds are distinguished: liquid clouds ($T > 273.15$ K), mixed-phase clouds ($238.15 \leq T \leq 273.15$ K) and ice clouds ($T < 238.15$ K), each using dedicated microphysical parameterizations. Formation of cloud droplets is described following Abdul-Razzak and Ghan (2000), calculating the fraction of activated aerosol particles at a given supersaturation as a function of their size and composition. Here, it is assumed that only the soluble compounds ($SO_4^{2-}$, $NO_3^-$, $NH_4^+$, $Na^+$ and $Cl^-$) contribute to the mean hygroscopicity parameter which controls the critical supersaturation for particle activation. An alternative formulation by Petters and Kreidenweis (2007) to calculate the supersaturation based on a single $\kappa$ parameter has also been implemented in EMAC and coupled to MADE3 in this work. Results of sensitivity simulations (see Sect. 4.3), however, revealed no significant differences in cloud droplet number concentration (CDNC) obtained with the different approaches and the Abdul-Razzak and Ghan (2000) approach is used to calculate the supersaturation in the simulation evaluated in this work.

Formation of ice crystals is described using different parameterizations for mixed-phase and cirrus clouds. In the mixed-phase regime, ice formation is assumed to occur via contact nucleation of dust particles and immersion freezing of BC and dust particles, according to the description by Lohmann and Diehl (2006) and Hoose et al. (2008). Dust is assumed to behave like a montmorillonite mineral in terms of its INP properties. Deposition nucleation of BC in the mixed-phase regime is considered to be negligible and not included. The Wegener-Bergeron-Findeisen process (Wegener, 1911; Bergeron, 1928; Findeisen, 1938) is parameterized according to Lohmann et al. (2007).

In the cirrus regime ($T < 238.15$ K), the parameterization by Kärcher et al. (2006) is used, which considers ice formation through the competition of various ice formation mechanisms for condensable water vapor: homogeneous freezing, deposition nucleation on BC and mineral dust, and immersion freezing on mineral dust, as well as the growth of pre-existing ice crystals. We note again that deposition nucleation of BC may in reality be pore condensation and freezing (Wagner et al., 2016; Marcolli,

2017; Mahrt et al., 2018; David et al., 2019). With respect to the original scheme by K14, in this work we further include black carbon as a potential ice-nucleating particle for heterogeneous freezing in cirrus clouds. In each of the heterogeneous freezing modes, the ice nucleation properties are described in the cloud scheme by means of two parameters: the active fraction $f_a$ of potential INPs that actually nucleate ice crystals and the critical supersaturation $S_c$ at which the freezing starts. The values assumed in this study for these parameters are summarized in Table 1: for deposition nucleation of insoluble dust and immersion freezing of coated (mixed) dust, we use the same values as K14, based on the laboratory studies by Möhler et al. (2006, 2008). For ice nucleation of BC we follow Hendricks et al. (2011), while further sensitivity experiments with varying ice nucleating properties for BC are planned in a follow-up study. The cirrus parameterization by Kärcher et al. (2006) implements the competition among the different ice formation processes in decreasing order of efficiency, i.e. from the heterogeneous freezing of the INPs with the lowest $S_c$ to homogeneous freezing. Due to a cooling of air parcels induced by updrafts, $S_c$ increases until the consumption of supersaturated water vapour by the growth of freshly formed or pre-existing ice crystals is large enough to terminate this process. The consumption of water vapour via depositional growth of pre-existing ice crystals is accounted for by reducing the vertical velocity in Eq. (1) by a so-called fictitious downdraft. If the cooling rate which corresponds to the reduced vertical velocity is still large enough to generate sufficiently high supersaturations, the heterogeneous and homogeneous ice formation processes, and the competition among them, can take place. Ice crystals larger than 200 $\mu$m in volume equivalent sphere diameter, typically formed by aggregation, are transferred to snow crystals which are assumed to be removed within one model time-step by precipitation, melting or sublimation (Levkov et al., 1992). This is introduced to avoid model instabilities which may arise due to a too fast sedimentation of large ice crystals (K14). Multiple ice modes (heterogeneous and homogeneous) are considered only for the ice nucleation and depositional growth processes, while for aggregation, accretion and transport a unimodal approach is used. The resulting ice crystal number concentration and ice water content are given by the sum of the concentrations in the individual ice modes. Further details on the cirrus parameterization, including the results of a box-model simulation, can be found in Kuebbeler (2013) and K14.

**Table 1.** Ice nucleation properties assumed for the different modes of heterogeneous ice formation in the cirrus scheme: ice active fraction $f_a$ and critical supersaturation $S_c$. $S_i$ is the ice supersaturation.

| Ice mode | | $S_c$ | $f_a$ | Reference |
|---|---|---|---|---|
| Dust deposition | $T \leq 220\text{K}$ | 1.1 | $\exp\left[2\left(S_i - S_c\right)\right] - 1$ | Kuebbeler et al. (2014) |
| | $T > 220\text{K}$ | 1.2 | $\exp\left[0.5\left(S_i - S_c\right)\right] - 1$ | |
| Dust immersion | | 1.3 | 0.05 | Kuebbeler et al. (2014) |
| BC | | 1.4 | 0.0025 | Hendricks et al. (2011) |

An important difference with respect to the original implementation of K14 in ECHAM5-HAM is introduced here. It concerns the calculation of the number concentrations of potential INPs for the different ice formation modes, which needs to be provided as an input to both the mixed-phase and the cirrus cloud parameterizations. The calculation of these parameters has

been completely revised here, to account for the structural differences between the HAM and MADE3 aerosol schemes, the latter providing a more detailed description of aerosol mixing states, as explained in the next subsection.

## 3.1 Calculation of the number concentration of potential INPs

As mentioned in Sect. 2, aerosol particles in MADE3 are distributed across three log-normal size modes (Aitken, accumulation and coarse mode), with three possible mixing states (soluble, insoluble, and mixed). Following the same notation as in K19, we indicate the MADE3 modes Aitken, accumulation and coarse with the index $k$, $a$ and $c$, respectively. Mixing states are indicated by $s$, $i$ and $m$ for soluble, insoluble and mixed, respectively. The relevant INPs considered in this study, namely BC and mineral dust (DU), are only present in the insoluble and mixed modes of MADE3, and mineral dust is only tracked in the accumulation and coarse mode. Of the 9 modes normally required in MADE3 for each aerosol compound, only 6 are hence needed for BC ($BC_{km}$, $BC_{ki}$, $BC_{am}$, $BC_{ai}$, $BC_{cm}$, and $BC_{ci}$) and 4 for mineral dust ($DU_{am}$, $DU_{ai}$, $DU_{cm}$, and $DU_{ci}$).

The number concentration of INPs available for contact freezing in the mixed-phase cloud regime and for deposition nucleation in the cirrus regime are indicated by $N^{cnt(mp)}$ and $N^{dep(c)}$, respectively. Deposition nucleation in mixed-phase clouds is neglected, since observations show that this process is probably not important for ice formation in mixed-phase clouds (Ansmann et al., 2008). In K14, the number of particles available for immersion freezing in mixed-phase clouds was estimated as a fraction of the number of aerosol particles activated to form cloud droplets $N^{act}$. However, this approach is not suitable for cirrus, where immersion freezing occurs in mixed solution aerosols, i.e. aerosol particles that underwent hygroscopic growth, rather than in cloud droplets. A different approach for calculating the number concentration of INPs available for immersion freezing in mixed-phase ($N^{imm(mp)}$) and cirrus clouds ($N^{imm(c)}$) is therefore introduced as part of this study. All the number concentrations calculated in this section are checked for consistency in the code, to make sure that the estimated number concentrations in each mode are not larger that the total number concentration in the mode itself. To simplify the notation, this check is not explicitly included in the equations below.

We first estimate the number concentration of dust particles in each mode, starting from dust mass concentration and using the conversion function

$$C_{DU}(D_j, \sigma_j) = \frac{6}{\pi} \frac{1}{D_j^3 \exp(4.5 \ln^2 \sigma_j) \rho}, \tag{2}$$

with $\rho$ = 2500 kg/m$^3$ for dust. $D_j$ and $\sigma_j$ are the lognormal size distribution parameters (median diameter and geometric standard deviation) of mode $j$, for which we follow the AeroCom recommendations (Dentener et al., 2006): $D_a = 0.42$ $\mu$m and $\sigma_a = 1.59$, for the accumulation mode, and $D_c = 1.30$ $\mu$m and $\sigma_c = 2.0$, for the coarse mode. The same parameters are also used to calculate the number of emitted dust particles in the model (see Sect. 2). This means that we are neglecting the aging of the size distribution due to dust-dust coagulation. This process has a limited efficiency due to the comparatively small number concentration of mineral dust particles (no dust is present in the Aitken mode given the typically large sizes of mineral dust particles). The number of BC particles is then estimated based on the total number of particles and the number of dust particles as described below.

Since no dust is present in the mixed and insoluble Aitken modes, each particle of these modes contains BC. Note that organic carbon cannot generate BC-free particles in these modes since it is assumed to be emitted internally mixed with BC in the form of "soot" (K19). Furthermore, only the mixed mode BC particles can be activated to form cloud droplets. For mixed-phase clouds, we therefore assume:

$$N_{\mathrm{BC,k}}^{\mathrm{imm(mp)}} = N_{\mathrm{km}}^{\mathrm{act}} \tag{3}$$

We do not include contact freezing of BC in the mixed-phase regime as its effect is considered largely uncertain (Lohmann and Hoose, 2009). For the cirrus regime, the number of potential Aitken-mode-sized INPs which could lead to immersion or deposition freezing coincides with the total number of particles in the mixed or insoluble Aitken mode, respectively:

$$N_{\mathrm{BC,k}}^{\mathrm{imm(c)}} = N_{\mathrm{km}} \qquad N_{\mathrm{BC,k}}^{\mathrm{dep(c)}} = N_{\mathrm{ki}} \tag{4}$$

In the mixed and insoluble accumulation modes, dust is present in the typical accumulation mode size (see above), since smaller dust particles are not considered and coarse dust particles can only reside in the coarse modes. In this case we estimate the number of dust particles from their mass $M$ using Eq. (2) and derive the number of potential dust INPs as follows:

$$N_{\mathrm{DU,a}}^{\mathrm{imm(mp)}} = M_{\mathrm{DU,am}}\, C_{\mathrm{DU,a}}\, \frac{N_{\mathrm{am}}^{\mathrm{act}}}{N_{\mathrm{am}}} \tag{5}$$

$$N_{\mathrm{DU,a}}^{\mathrm{imm(c)}} = M_{\mathrm{DU,am}}\, C_{\mathrm{DU,a}} \tag{6}$$

$$N_{\mathrm{DU,a}}^{\mathrm{cnt(mp)}} = M_{\mathrm{DU,ai}}\, C_{\mathrm{DU,a}} \qquad N_{\mathrm{DU,a}}^{\mathrm{dep(c)}} = M_{\mathrm{DU,ai}}\, C_{\mathrm{DU,a}} \tag{7}$$

It cannot be excluded that these potential dust INPs also contain BC as a consequence of coagulation. Due to the large size of the dust particles compared to BC, we assume however that dust dominates the ice nucleation properties of the particles. The remaining number of particles in the insoluble and mixed accumulation modes can then be ascribed to soot particles (internally mixed black and organic carbon):

$$N_{\mathrm{BC,a}}^{\mathrm{imm(mp)}} = \max(0, N_{\mathrm{am}}^{\mathrm{act}} - N_{\mathrm{DU,a}}^{\mathrm{imm(mp)}}) \tag{8}$$

$$N_{\mathrm{BC,a}}^{\mathrm{imm(c)}} = \max(0, N_{\mathrm{am}} - N_{\mathrm{DU,a}}^{\mathrm{imm(c)}}) \tag{9}$$

$$N_{\mathrm{BC,a}}^{\mathrm{dep(c)}} = \max(0, N_{\mathrm{ai}} - N_{\mathrm{DU,a}}^{\mathrm{dep(c)}}) \tag{10}$$

The insoluble coarse mode is dominated by dust, since it is unlikely that self-coagulation of insoluble accumulation mode BC particles leads to growth into the insoluble coarse mode (BC mass is limited and the self-coagulation frequency is comparatively low). Hence, coarse dust particles are needed to form this mode. This results in:

$$N_{\mathrm{DU,c}}^{\mathrm{cnt(mp)}} = N_{\mathrm{ci}} \qquad N_{\mathrm{DU,c}}^{\mathrm{dep(c)}} = N_{\mathrm{ci}} \tag{11}$$

$$N_{\mathrm{BC,c}}^{\mathrm{dep(c)}} = 0 \tag{12}$$

In the mixed coarse mode the mixing state is uncertain, since particles can be composed of dust from both the accumulation and the coarse size range, whose relative contribution is not known. Thus mass-to-number conversion is not as straightforward as in the accumulation mode case. We need to distinguish two cases, based on the relative abundance of dust particles. We define the dust number fraction in this mode as:

$$f_{\mathrm{DU}} = \frac{M_{\mathrm{DU,cm}} C_{\mathrm{DU,c}}}{N_{\mathrm{cm}}} \tag{13}$$

We use the conversion factor $C_{\mathrm{DU,c}}$ of coarse dust particles, given in Eq. (2) to estimate the number fraction, since these particles dominate the dust mass (possible mass contributions of accumulation mode dust are small, according to Dentener et al., 2006). Hence, an estimate of the coarse dust particle number based on the total dust mass in the mode appears to be a good approximation. It provides a minimum estimate of the number of dust containing particles in the mode, since also many accumulation-mode-sized dust particles might be present in the mode due to coagulation. For dust-dominated regimes, e.g. at or in the vicinity of deserts, it can be expected that $f_{\mathrm{DU}}$ is large and that also the non-coarse-dust particles in the mode contain many accumulation-mode-sized dust immersions. It can also be expected that BC has a comparatively small contribution under these conditions. Hence, all particles of the mode can be regarded as possible dust ice-nucleating particles. In the present study we assume that mineral dust dominates in the mode where $f_{\mathrm{DU}} \geq 0.7$. In this case, the above assumptions result in:

$$N_{\mathrm{DU,c}}^{\mathrm{imm(mp)}} = N_{\mathrm{cm}}^{\mathrm{act}} \tag{14}$$

$$N_{\mathrm{DU,c}}^{\mathrm{imm(c)}} = N_{\mathrm{cm}} \tag{15}$$

$$N_{\mathrm{BC,c}}^{\mathrm{imm(mp)}} = 0 \qquad N_{\mathrm{BC,c}}^{\mathrm{imm(c)}} = 0 \tag{16}$$

If $f_{\mathrm{DU}} < 0.7$ we assume that BC plays a major role and that the minimum estimate of the number of dust containing particles applies. This results in:

$$N_{\mathrm{DU,c}}^{\mathrm{imm(mp)}} = M_{\mathrm{DU,cm}} C_{\mathrm{DU,c}} \frac{N_{\mathrm{cm}}^{\mathrm{act}}}{N_{\mathrm{cm}}} \tag{17}$$

$$N_{\mathrm{DU,c}}^{\mathrm{imm(c)}} = M_{\mathrm{DU,cm}} C_{\mathrm{DU,c}} \tag{18}$$

$$N_{\mathrm{BC,c}}^{\mathrm{imm(mp)}} = \max(0, N_{\mathrm{cm}}^{\mathrm{act}} - N_{\mathrm{DU,c}}^{\mathrm{imm(mp)}}) \tag{19}$$

$$N_{\mathrm{BC,c}}^{\mathrm{imm(c)}} = \max(0, N_{\mathrm{cm}} - N_{\mathrm{DU,c}}^{\mathrm{imm(c)}}). \tag{20}$$

Despite the admittedly many assumptions required to estimate the number of coarse immersion INPs, we note that the resulting uncertainties are probably small, since the contribution of coarse particles to the number concentration of INPs is mostly small compared to the corresponding contribution of the accumulation mode. Sensitivity studies show little to no variation in ice water content and ice crystal number concentration for values of $f_{DU}$ ranging from 0.6 to 0.9.

## 4 Model evaluation

In this section we evaluate the performance of EMAC-MADE3 in the coupled configuration. In the context of this study, the coupling refers to the explicit simulation of the aerosol-cloud and aerosol-radiation interactions by the model. The representation of aerosol quantities such as particle mass and number concentrations, size distributions as well as aerosol optical properties was extensively evaluated in K19 against a comprehensive set of observational data from different sources. In K19, we concluded that MADE3 is able to capture the global pattern of aerosol mass and number distribution with deviations which are in line with the results of other global aerosol models available in the literature. The conclusions of the K19 evaluation on the aerosol representation in the uncoupled model version still hold for the aerosol-climate coupled version discussed here, since the aerosol-cloud and aerosol-radiation couplings do not lead to significant changes in the global aerosol characteristics. However, the present configuration uses a higher vertical resolution than in K19, with 41 instead of 19 vertical levels. This leads to some differences in the representation of aerosol in the cirrus-relevant upper tropospheric regions, but showing in most cases a slightly improved model performance in these regions (see Figs. S1 and S2 in the Supplement). In this study, we focus on cloud and radiation variables and analyze the performance of EMAC-MADE3 in reproducing essential quantities (such as total cloud cover, cloud liquid and ice water, cloud droplet and ice crystal number concentrations, and cloud radiative effects) compared to satellite and in situ observations. Since the present configuration is developed with a specific focus on cirrus clouds, a special attention will be devoted to this aspect.

The observational datasets used for tuning the model are summarized in Table 2 and further details are provided in the respective sections for each variable (Sects. 4.2–4.6). To allow for a direct comparison between model and observations, satellite data are regridded to the EMAC $2.8° \times 2.8°$ horizontal grid and are compared on a monthly-climatology basis. The in situ data for cirrus clouds are not provided on a standard latitude-longitude grid, but as probability distribution functions in 1-K temperature bins. In this case, the model output is sampled in the same bins as the observations in order to generate a comparable distribution. When possible, the observational time periods are chosen to match the simulated one (1996–2005).

### 4.1 Model tuning in comparison to observations

Following a similar approach as in Lohmann and Ferrachat (2010), we first analyse how sensitive the model performance in representing key cloud and radiation variables is when varying certain tuning parameters. We recall that the EMAC-MADE3 configuration being tuned in this work is nudged, i.e. meteorological variables such as temperature, winds, and the logarithm of surface pressure are relaxed towards reanalysis data. In line with the designed application target of this version of the model, this allows to run different simulations (such as perturbation experiments) with very similar meteorological conditions.

**Table 2.** Summary of the observational dataset used for tuning cloud and radiation variables in EMAC-MADE3.

| Variable | Dataset | Type | Temporal coverage | Reference |
|----------|---------|------|-------------------|-----------|
| Cloud cover | ESACCI-CLOUD v3.0 | Satellite | 1996–2005 | Stengel et al. (2017) |
| Liquid water path | MAC | Satellite | 1996–2005 | Elsaesser et al. (2017) |
| Cloud droplet number concentration | Bennartz17 | Satellite | 2003–2015 | Bennartz and Rausch (2017) |
| Ice water content | Krämer16 | In situ | 1999–2014 | Krämer et al. (2009, 2016) |
| Ice crystal number concentration | Krämer16 | In situ | 1999–2014 | Krämer et al. (2009, 2016) |
| Precipitation | GPCP-SG v2.3 | Satellite | 1996–2005 | Adler et al. (2018) |
| Cloud radiative effects | CERES-EBAF v4.0 | Satellite | 2001–2010 | Loeb et al. (2018) |

Such a model setup is most suitable for short-term time-slice experiments, that aim at isolating the effects of specific sources and processes, which would be statistically and numerically far more challenging without nudging. The use of the nudging technique has to be kept in mind while tuning the model, since nudging unavoidably impacts on the model climate, as it introduces a forcing component by modifying the model's temperature profile, which in turn perturbs the radiative balance

(Zhang et al., 2014). A previous study by Schultz et al. (2018) based on the ECHAM6 model showed that temperature nudging may introduce a radiative imbalance around $5 \, \mathrm{W \, m^{-2}}$ with respect to an otherwise identical configuration without nudging. In the following, we will apply our tuning procedure to the nudged setup. Once the optimal configuration is identified, we will additionally perform a control experiment in free running mode to address the above issue and quantify the actual impact of nudging on the radiative balance in the tuned model configuration.

Our tuning approach focuses in particular on the enhancement factor of the rate of rain formation by autoconversion $\gamma_r$, the rate of snow formation by aggregation $\gamma_s$, the minimum cloud droplet number concentration $\mathrm{CDNC_{min}}$, and the size of newly nucleated aerosol particles $d_{\mathrm{nuc}}$. The minimum CDNC is introduced in the model to avoid unrealistically low concentrations of cloud droplets in pristine conditions. The parameter $d_{\mathrm{nuc}}$ is used to describe the initial growth of freshly formed sulfuric acid water clusters into larger sulfate aerosol particles. Since such nucleation and growth events frequently occur on spatial

scales which cannot be resolved by the global model, the use of this parameter enables the implicit consideration of these subgrid-scale processes. In K19 a value $d_{\mathrm{nuc}} = 10$ nm was chosen, motivated by a better agreement of simulated number concentrations with observations and supported by new particle formation measurements. Here we explore how this parameter can also affect cloud and radiation variables. Lohmann and Ferrachat (2010) further considered the inhomogeneity factor of ice clouds and the entrainment rate for deep convection as tuning parameters, which in our configuration are set to 0.85

and $10^{-4} \, \mathrm{kg \, m^{-3} \, s^{-1}}$, respectively, but their variation is not further explored. We tested 5 values for each of the 4 tuning parameters $\gamma_r$, $\gamma_s$, $\mathrm{CDNC_{min}}$ and $d_{\mathrm{nuc}}$, varying across a range of approximately one order of magnitude. We then calculated their effect on six cloud variables (total cloud cover, liquid water path (LWP) over the oceans, CDNC over the oceans, ice water content in cirrus clouds ($\mathrm{IWC_{cirrus}}$), ice crystal number concentration in cirrus clouds ($\mathrm{ICNC_{cirrus}}$), and precipitation) and three radiation variables (shortwave and longwave cloud radiative effects (SWCRE and LWCRE, as the difference between

all-sky and clear-sky radiation fields), as well as the net radiative balance). Note that exploring the full parameter space, i.e. all possible combinations of the 5 values for the 4 tuning parameters, is not feasible as it would require performing $5^4 = 625$ model simulations. So we only explore the model sensitivity for each single parameter while keeping the others fixed at a reference value, which corresponds to the median of the explored range, i.e. $\gamma_r = 8$, $\gamma_s = 800$, $CDNC_{min} = 10\ cm^{-3}$, and $d_{nuc} = 10$ nm. This limits the numbers of simulations to be performed to 17 (i.e., $5 \times 4 = 20$ simulations minus 3 redundant cases). To further reduce the computational costs, the tuning simulations cover a period of only three years (1999–2001). To quantitatively characterize the impact of the 4 tuning parameters on the model variables, the normalized root mean square error (NRMSE) of the model with respect to the observations is calculated for each variable-parameter combination:

$$\text{NRMSE} = \sqrt{\frac{\sum_i (M_i - O_i)^2}{n}} \bigg/ \frac{\sum_i O_i}{n} \tag{21}$$

where $M_i$ and $O_i$ represent the model and observational data, respectively. When comparing model and satellite data, the index $i$ runs across all model grid boxes and the 12 timesteps resulting from the calculation of a monthly climatology. For the in situ data, the index $i$ corresponds to the median value of the distribution for each 1-K temperature bins. In the case of gridded data, a weighting factor proportional to the grid box area and to the length of each month is also applied in Eq. (21).

The results of this analysis are summarized in Fig. 1: each panel depicts the variation of a given variable over the range of values for the tuning parameter, the black circles marking the corresponding NRMSE values (note that for the radiative balance the globally-averaged value is shown). To quantitatively describe this variation, each panel is color-coded according to the relative standard deviation of the NRMSE ($RSD = \sigma_{NRMSE} / \overline{NRMSE}$) for the variable-parameter combination shown in that panel. This helps to identify the combinations which display a low ($RSD \leq 0.1$), medium ($0.1 < RSD \leq 0.2$) or high ($RSD > 0.2$) variation with the value of tuning parameters.

The autoconversion rate (first column of Fig. 1) controls the removal of liquid water from the clouds via precipitation and has therefore an impact on LWP in the model, also changing the radiative balance, while the effect on the other variables is less significant ($RSD \leq 0.1$). Choosing a high autoconversion rate minimises the NRMSE of LWP, but at the expense of a large radiative imbalance. A good compromise is $\gamma_r = 5$, which limits the imbalance to 6.2 W m$^{-2}$ while keeping the NRMSE of LWP reasonably low. Note that for values $\gamma_r < 5$, the NRMSE of LWP grows rapidly, so this is the lowest among the explored $\gamma_r$ values for which a reasonable deviation in both affected variables can be attained. Similarly, the aggregation rate (second column) controls the conversion of ice crystals to snow in ice clouds and has consequently a large ($RSD > 0.2$) impact on IWC and ICNC in cirrus clouds, while its influence on the radiative balance is less pronounced than for the autoconversion rate. In this case, however, the two mostly affected variables behave similarly when varying $\gamma_s$, with a decreasing NRMSE for increasing values of $\gamma_s$, so that setting it to the maximum of the investigated range, $\gamma_s = 1300$, seems to be the most appropriate choice. The minimum CDNC (third column) slightly impacts the NRMSE of $ICNC_{cirrus}$ and has an important effect on the radiative balance in the model. In ECHAM5-HAM and ECHAM6-HAM, this parameter was varied between 10 and 40 cm$^{-3}$ (Lohmann and Ferrachat, 2010; Neubauer et al., 2019). A high value would help to minimise both the NRMSE of $ICNC_{cirrus}$ and the radiative imbalance in EMAC-MADE3, but lower values for this threshold parameter are more consistent with the observations in pristine marine regions (Bennartz, 2007; Karydis et al., 2011; Bennartz and Rausch, 2017). Here, therefore,

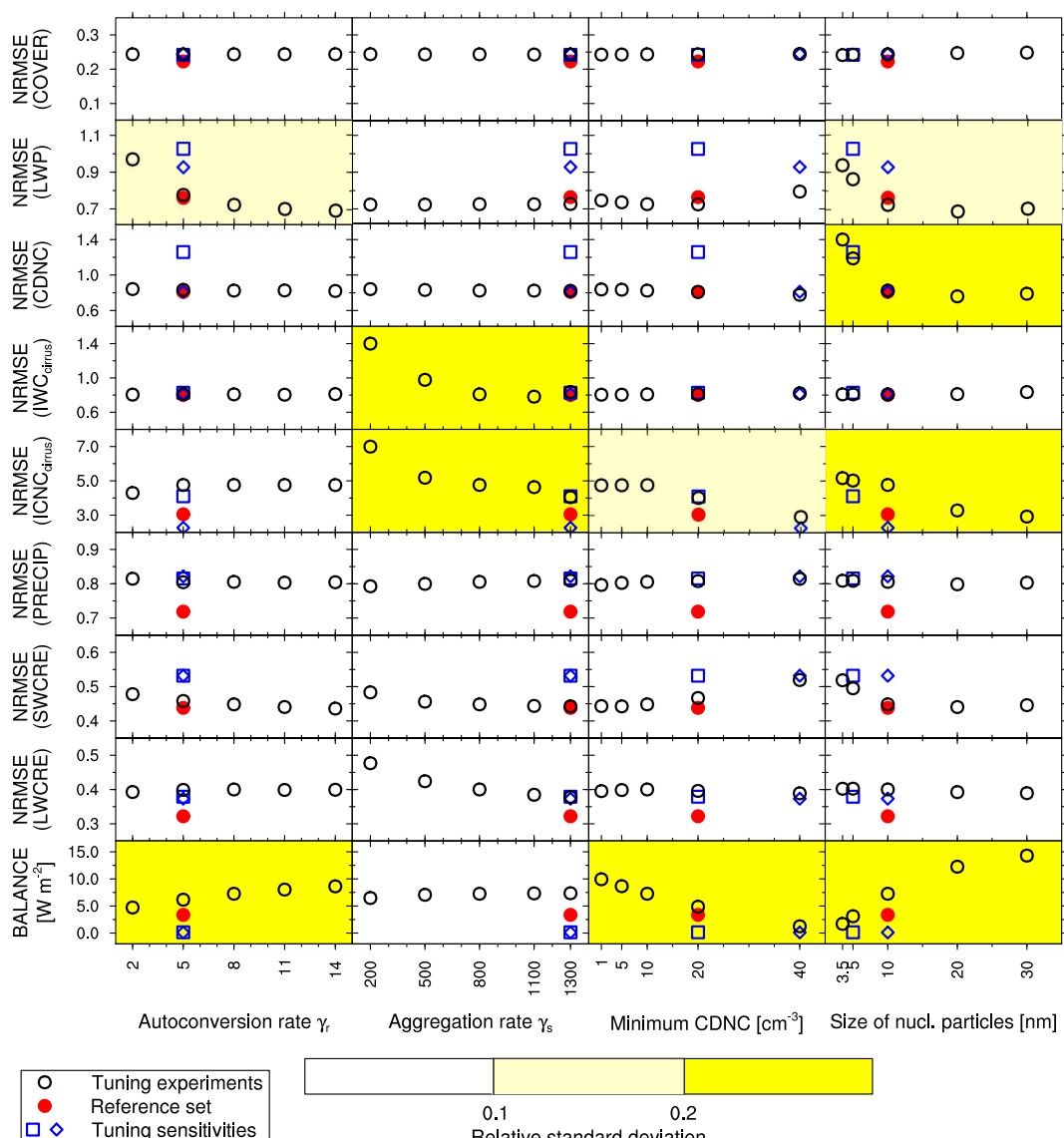

**Figure 1.** NRMSE of selected cloud and radiation variables in the tuning experiments, together with the resulting model top-of-the-atmosphere radiative balance (in W m$^{-2}$). Each column represents a tuning parameter and shows the NRMSE for the five tuning simulations (black circles) performed to explore the effects of its variation, while keeping the others fixed at their central value (note that for the radiative balance the actual value and not the NRMSE is shown). The red circle represent the results for the reference set of tuning parameters ($\gamma_r = 5$, $\gamma_s = 1300$, $CDNC_{min} = 20$ cm$^{-3}$, and $d_{nuc} = 10$ nm), while the blue square and diamond show the results of two additional sensitivity runs to further explore the variation on $CDNC_{min}$ and $d_{nuc}$. The NRMSE is calculated with respect to the observations given in Table 2. The yellow shadings represent the relative standard deviation of the NRMSE within each panel. For SWCRE, the absolute value of the NRMSE is shown for more clarity (since it is negative via the normalization).

we choose $CDNC_{min} = 20$ cm$^{-3}$, to avoid tuning the model using unrealistic values, but at the price of a slightly worse representation of ICNC in cirrus and a slightly higher radiative imbalance. The size of newly nucleated particles $d_{nuc}$ (fourth column) has a significant impact on aerosol number concentrations, as shown by K19. Since aerosol particles serve as cloud condensation nuclei for cloud droplet formation, this parameter primarily controls CDNC in the model, hence impacting LWP,
and $ICNC_{cirrus}$, with a consequent effect on the radiative balance. As for the autoconversion rate, the tuning experiments reveal some trade-offs between these variables: large values of $d_{nuc}$ allow to minimise the NRMSE of LWP, $ICNC_{cirrus}$ and (to a lesser extent) CDNC, but again at the expenses of the radiative balance, which grows rapidly over 10 W m$^{-2}$ for $d_{nuc} > 10$ nm. A good compromise is $d_{nuc} = 10$ nm, which allows to keep the NRMSE of CDNC low and is also supported by K19, who found a good agreement with aerosol number concentration measurements when choosing $d_{nuc} = 10$ nm.

In summary, this choice of the four parameters ($\gamma_r = 5$, $\gamma_r = 1300$, $CDNC_{min} = 20$ cm$^{-3}$, and $d_{nuc} = 10$ nm) represents our reference set and is used to perform a full ten-year simulation which is then evaluated in more detail in the next section. The NRMSE resulting with this choice of the tuning parameters is shown with the red circles in Fig. 1. Note that this set of the tuning parameters does not match any of the other sets marked with the black circles. This is because, as explained above, the tuning strategy adopted in this study aims at exploring the model's sensitivity to a set of tuning parameters, but it is
limited to single-dimensional variation studies, in which the value of a single parameter is changed while the others are kept fix at a reference value. Deciding on a reference set of tuning parameters always involves expert judgment and a compromise between certain basic principles (Schmidt et al., 2017), as the choice of priorities must be guided by the main application target of the model setup to be tuned. If, for instance, the model is to be coupled to an interactive ocean, a close-to-zero radiation balance at the top of the atmosphere must be the central goal of tuning. In this study, the first priority has been laid on providing
a reasonably good agreement with the observed values of the main cloud and radiation variables, although the model retains a radiative imbalance of 3.4 W m$^{-2}$. A lower imbalance could be obtained choosing a different set of tuning parameters, but at the expense of a worse agreement with the observations of other essential variables. This has been tested by performing two additional tuning simulations, increasing $CDNC_{min}$ to 40 cm$^{-3}$ and reducing $d_{nuc}$ to 5 nm with respect to the reference set, respectively. The results, shown by the blue markers in Fig. 1, demonstrate that this choice would improve the radiative balance
to a value of 0.1 W m$^{-2}$ in both cases, but would also lead to larger errors in most of the other variables. As discussed above, however, a significant imbalance is introduced by temperature nudging, which is applied in all experiments shown here. To quantify this, a simulation with the tuned configuration but in free running mode is performed, which encouragingly results in a radiative balance of $-0.5$ W m$^{-2}$. Hence, while the nudged configuration – with its imbalance of 3.4 W m$^{-2}$ – is well suited for its designed purpose, also the free running model would meet the common requirements. We stress again that introducing
a change of a few W m$^{-2}$ on the radiative balance through nudging is fully consistent with the study by Schultz et al. (2018) mentioned above.

Table 3 summarizes the results of the tuned simulation, compared with similar experiments performed with ECHAM5-HAM by K14, ECHAM6-HAM2 by Lohmann and Neubauer (2018), EMAC-GMXe by Bacer et al. (2018), NCAR-CAM5.3 by Penner et al. (2018) and ECHAM6.3-HAM2.3 by Neubauer et al. (2019). The performance of the EMAC-MADE3 coupled
configuration is in line with these models and mostly close to the observed values given in the table.

In the rest of this section, we extend the analysis by further evaluating the mean state of cloud and radiation variables against satellite data using the diagnostics included in the ESMValTool v1.1.0 (Eyring et al., 2016). Most of these observational data sets have already been used for the model tuning procedure described above, but they are further analysed to assess the performance of the tuned model configuration in representing specific aspects of the target variables and better quantify the remaining biases, by looking, for instance, at their spatial distribution. However, we complement these datasets by additional ones to provide a more robust and independent evaluation.

**Table 3.** Summary of the globally-averaged cloud and radiation variables obtained with the reference set of tuning parameters ($\gamma_r = 5$, $\gamma_s = 1300$, $\text{CDNC}_{\text{min}} = 20\,\text{cm}^{-3}$, and $d_{\text{nuc}} = 10\,\text{nm}$), compared with the observations summarized in Table 2, and with the results of other global models: ECHAM5-HAM (Kuebbeler et al., 2014), ECHAM6-HAM2 (Lohmann and Neubauer, 2018), EMAC-GMXe (Bacer et al., 2018), NCAR-CAM5.3 (Penner et al., 2018) and ECHAM6.3-HAM2.3 (Neubauer et al., 2019). The uncertainty ranges in the observations of cloud cover, LWP, and CDNC are calculated from the standard errors provided in the respective datasets; for precipitation the uncertainty is taken from Adler et al. (2018); for $\text{IWC}_{\text{cirrus}}$ and $\text{ICNC}_{\text{cirrus}}$ the given ranges correspond to the 25/75% quantiles of the in situ measurements (averaged over the reported temperature range); for SWCRE and LWCRE they are taken from Loeb et al. (2018).

| | This study | Observations | ECHAM5-HAM | ECHAM6-HAM2 | EMAC-GMXe | NCAR-CAM5.3 | ECHAM6.3-HAM2.3 |
|---|---|---|---|---|---|---|---|
| Cloud cover [%] | 66.0 | 64.5±17.4 | 62.3 | 68.1 | [69.0; 70.0] | [69.3; 72.2] | [64; 69] |
| LWP oceans [$\text{g m}^{-2}$] | 84.1 | 83.0±10.2 | 55.6 | 70.6 | [72.7; 76.6] | [45.7; 57.7] | [71; 94] |
| CDNC [$\text{cm}^{-3}$] | 89.9 | 74.0±41.1 | – | – | – | – | [76, 80] |
| $\text{IWC}_{\text{cirrus}}$ [ppmv] | 5.7 | 7.2 [1.7; 29.2] | – | – | – | – | – |
| $\text{ICNC}_{\text{cirrus}}$ [$\text{cm}^{-3}$] | 0.08 | 0.03 [0.006; 0.10] | – | – | – | – | – |
| Precipitation [$\text{mm d}^{-1}$] | 3.1 | 2.7±0.2 | 2.87 | 2.99 | [2.89; 3.03] | [2.73; 2.80] | 3.0 |
| SWCRE [$\text{W m}^{-2}$] | −53.1 | −45.9±5.5 | −54.8 | −49.9 | [−58.1; −54.8] | [−66.3; −58.5] | [−53; −50] |
| LWCRE [$\text{W m}^{-2}$] | 27.4 | 28.1±4.4 | 28.8 | 24.1 | [28.9; 34.4] | [32.1; 36.7] | [24; 28] |
| Radiative balance [$\text{W m}^{-2}$] | 3.4 | – | −0.6 | – | [1.53; 4.65] | – | [−0.1; 0.4] |

## 4.2  Total cloud cover and cloud liquid water

In the top row of Fig. 2, multi-year average total cloud cover over the simulated time period (1996–2005) is compared with the ESA Climate Change Initiative (ESACCI) CLOUD satellite product, which is based on data from the passive imager sensors AVHRR, MODIS, ATSR-2, AATSR and MERIS (Stengel et al., 2017). The overall pattern is very well reproduced by EMAC, with a small positive bias in the tropics and a negative bias in the stratocumulus regions off the coasts of South America and Africa. These features are quite common in many global models, e.g. those participating in the CMIP3 and CMIP5 intercomparisons (Lauer and Hamilton, 2013), both in the AMIP and in the ocean-coupled configuration. Larger deviations between EMAC and the observations are found in polar regions, where, however, observational uncertainties are also larger (Lauer et al., 2017). Note that total cloud cover is only weakly controlled by the specific coupling evaluated here and is rather

a general feature of the core model, as demonstrated by the similar biases found by other studies using ECHAM5 and the Sundqvist et al. (1989) cloud cover scheme, such as Lohmann et al. (2007) and K14.

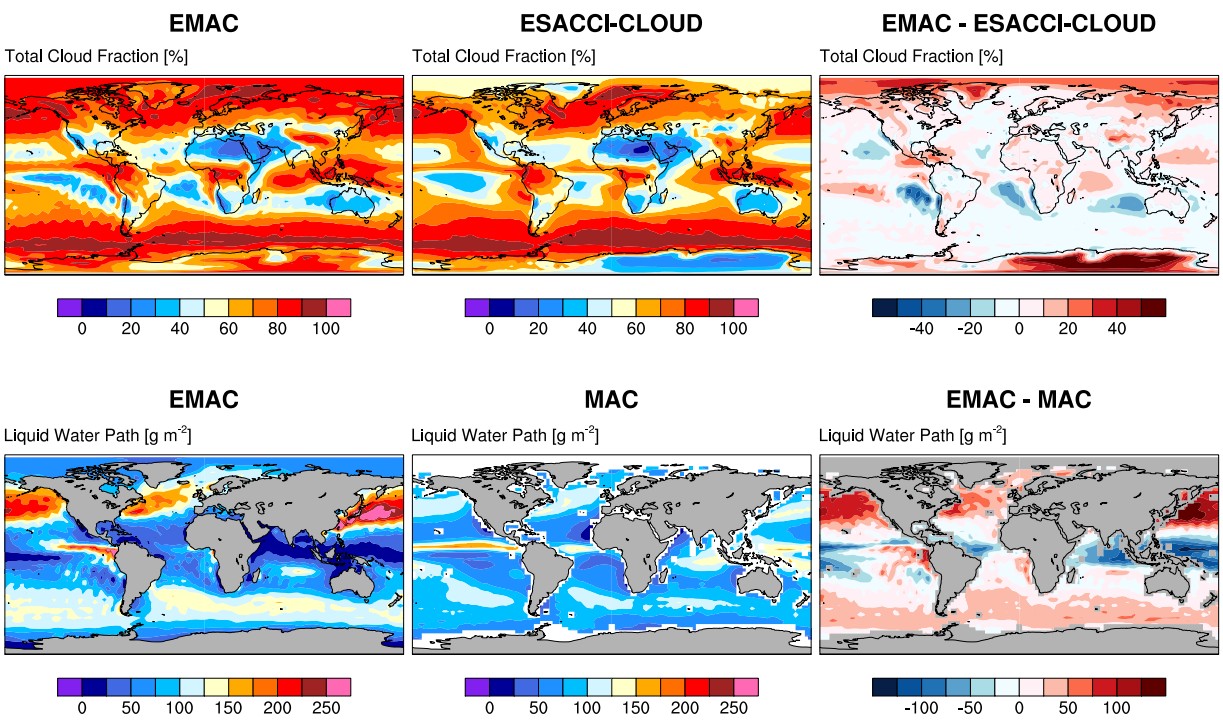

**Figure 2.** Multi-year (1996–2005) average total cloud cover (top row) and liquid water path (bottom row) simulated by EMAC (left), in the satellite data (middle), and the difference between model and observations (right).

    In the second row of Fig. 2, an analogous comparison is shown for LWP over the oceans against the Multisensor Advanced Climatology (MAC, Elsaesser et al., 2017), which combines data from different satellite sources over the ocean, including

5  SSM/I, TMI, AMSR-E, WindSat, SSMIS, AMSR-2, and GMI. Although the general pattern of LWP is reproduced by EMAC, several features are not consistent with observations: in particular, EMAC tends to simulate a higher LWP in the northern extra tropics, with particularly high values in the Western Pacific, and a lower LWP in the tropics, while it agrees better with the observations in the southern extra tropics. Most striking is the high bias in the Northwest Pacific ocean, which may be related to a high bias in the cloud lifetime in this region. As it will be shown in the next section, CDNC is also biased high in this region

10  in comparison to satellite data, which could in turn be the effect of a too high concentration of cloud condensation nuclei. These biases could also be partly related to the tendency of EMAC to underestimate low cloud fraction in the tropics and overestimate it in the extra-tropics (Räisänen and Järvinen, 2010; Righi et al., 2015a). Uncertainties in the prescribed emission fluxes could also contribute to these biases, especially in East Asia, where anthropogenic emissions in the year 2000 are higher than in other regions of the world and have further increased since then. As for the total cloud cover, similar deviations were found by

15  Lauer and Hamilton (2013) in the CMIP5 multi-model mean, which is characterized by large biases in the same regions.

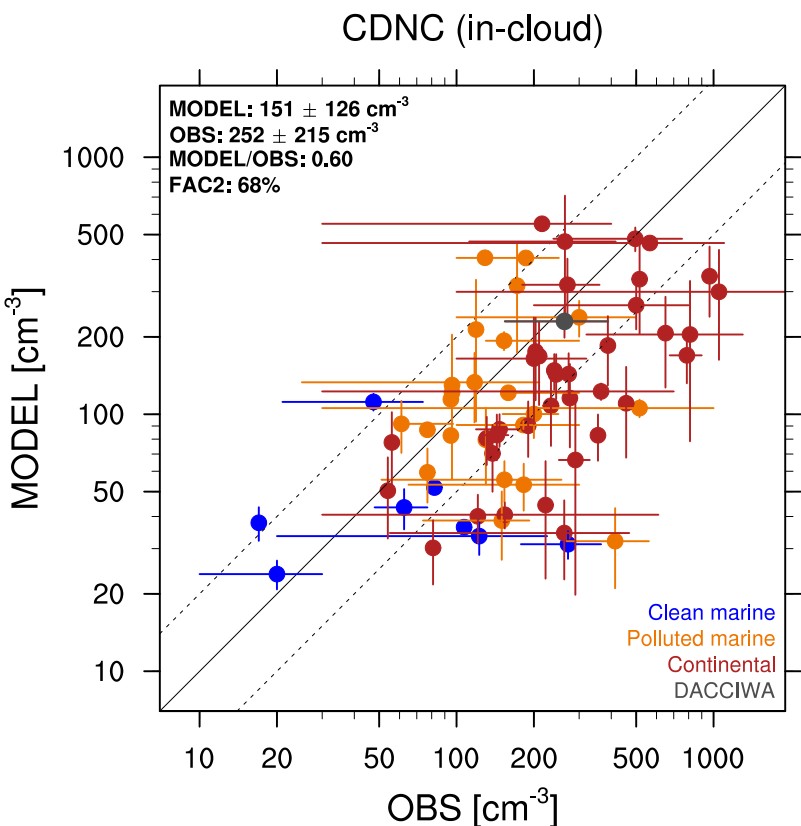

**Figure 3.** Scatterplot of simulated (vertical) vs. observed (horizontal) CDNC based on various satellite and in situ measurements collected by Karydis et al. (2011, 2017). Different colors represent different groups of measurement locations: clean marine (blue), polluted marine (orange) and continental (red). A further comparison with the mean of 12 flights performed during the DACCIWA campaign (Flamant et al., 2018) is shown in gray. The horizontal bars represent the range of reported values, while for the DACCIWA campaign it spans the range of the means from all flights. The vertical bars show the standard deviation in the model interannual variability. The mean and standard deviation of model and observations is shown on the top-left, together with their ratio and the percentage of points within a factor of 2 of the observations (FAC2, i.e. factors between 0.5 and 2), also indicated by the dashed lines.

### 4.3 Cloud droplet number concentration

We evaluate CDNC using a compilation of in situ measurements provided by Karydis et al. (2011, 2017), integrated with the measurements performed during 12 research flights during the DACCIWA campaign (Dynamics-Aerosol-Chemistry-Cloud Interactions in West Africa; Flamant et al., 2018; Taylor et al., 2019) in summer 2016 around Lomé (Togo). For this comparison, the model data are spatially colocated with the observations using a nearest-neighbor selection method for both the horizontal and the vertical coordinates. In the vertical, this is realized using the information provided for each location: either altitude (geopotential height in the model), pressure level, surface level (the lowermost hybrid model layer), or selecting the levels within the boundary layer as calculated by the model. The time selection is performed in a climatological way, by sampling the model output at a 5-hourly frequency for the same month(s) or season(s) as reported by each measurement, and averaging the selected time-steps over the whole (10 years) simulation period. For comparison with the DACCIWA data, the model output is further filtered, by selecting only the cloud scenes with a liquid water content above $0.01 \, \mathrm{g \, m^{-3}}$, which is the same criterion adopted in the measurements. Reported concentrations correspond to in-cloud values.

The result of this comparison is depicted as a scatterplot in Fig. 3: considering the observational uncertainties, EMAC simulates CDNC within a factor of two (i.e., factors in the range from 0.5 to 2) of the observations in most of the regions (i.e., 68% of cases), but it generally tends to underestimate this quantity. This is in line with recent results by Rothenberg et al. (2018), who used the MARC global aerosol model and applied various cloud droplet activation schemes. Their global integrated CDNC in the range $60\text{–}91 \, \mathrm{cm^{-3}}$ is lower than the average value simulated by EMAC ($151 \, \mathrm{cm^{-3}}$), which is closer to the range $75\text{–}135 \, \mathrm{cm^{-3}}$ reported by Penner et al. (2006) for three models also using the Abdul-Razzak and Ghan (2000) parameterization for cloud droplet activation.

We also compare the simulated CDNC with satellite retrievals, which provide a unique global picture of this quantity, although this kind of retrievals are still affected by considerable uncertainties (Grosvenor et al., 2018). Here we use a recent 13-year climatology by Bennartz and Rausch (2017), based on MODIS-AQUA retrievals. In-cloud CDNC, as reported in the observational dataset, are extracted from the model output with the same method used for comparing with in situ data, but considering CDNC at cloud top, as observed by the satellite, i.e. by taking the CDNC in the highest model level with a liquid cloud. An alternative method, taking the average CDNC through the cloudy part of the column, provides very similar results (not shown). This is expected, since the representation of liquid cloud formation in the EMAC cloud scheme follows the adiabatic parcel theory, assuming that newly formed cloud droplets at the cloud base are equally distributed in the vertical by mixing, regardless of the aerosol concentrations. An identical assumption is also done in the retrieval process by Bennartz and Rausch (2017). The results of this comparison are depicted in Fig. 4. In the Northern Hemisphere, EMAC captures the major spots of high CDNC over the Atlantic Ocean eastward of Canada and USA, over the Mediterranean and eastward of China, albeit with about 50% higher CDNC than MODIS. These spots also have a wider horizontal extent over the oceans than in the observations. This could be due to the generally high CDNC over the continents (as shown by the in situ data in Fig. 3) being too efficiently advected over the oceans or, as mentioned in Sect. 4.2, to a bias in the prescribed emissions, causing a too high aerosol concentration and hence a too high number of cloud condensation nuclei being activated. Another major bias is found

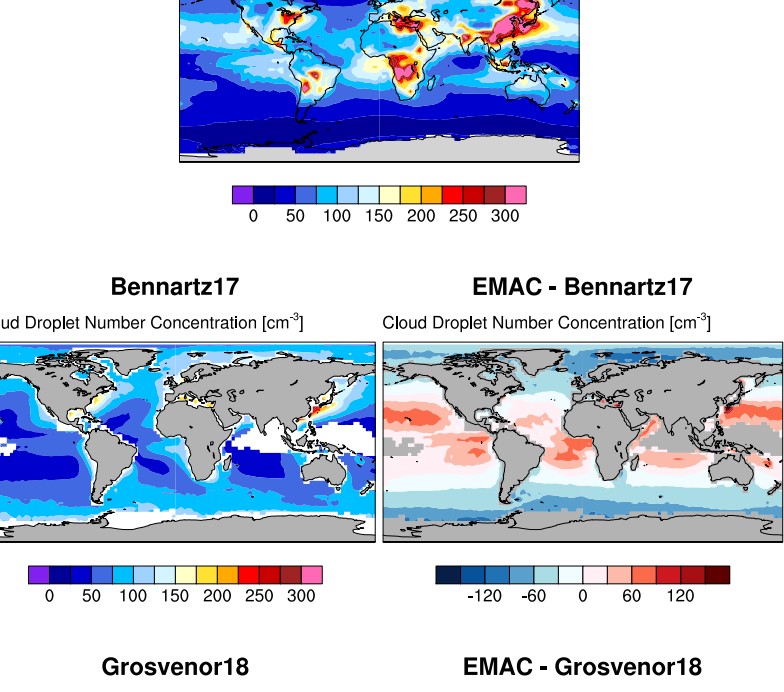

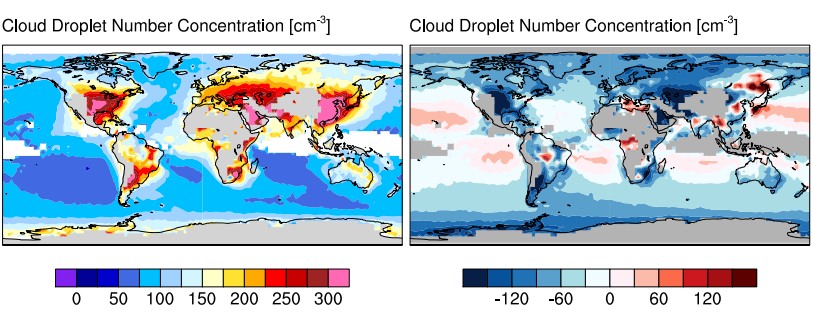

**Figure 4.** Multi-year (1996–2005) average of CDCN at cloud top, as simulated by EMAC (top) and observed by two satellite retrievals (middle row), as well as the difference between model and observations (bottom row).

at the Equator westward of central Africa, which could be due to biomass burning aerosol being transported over the Atlantic. Here again uncertainties in emissions of aerosols and precursor species could play an important role in explaining such bias. In addition to deficiencies in the aerosol representation, misrepresentations of the model dynamics could also explain deviations from the observations. Another source of uncertainty is related to the retrieval errors in the MODIS products for effective ra-
dius and optical depth, which are used to derive CDNC. According to Bennartz and Rausch (2017), this uncertainty is around 30% in the stratocumulus regions and 60–80% in the storm tracks, increasing towards the poles. Furthermore, this dataset is considered as highly uncertain for latitudes above 60° (Mulcahy et al., 2018; Rothenberg et al., 2018). To better characterize the observational uncertainties, we compare CDNC with another 13-year MODIS-based dataset (Fig. 4), namely the climatology by Grosvenor and Wood (2018) based on the retrieval method of Grosvenor and Wood (2014). The large positive bias of
EMAC-MADE3 over the tropical oceans is significantly lower when comparing with Grosvenor and Wood (2018) than with Bennartz and Rausch (2017), while the negative bias over the extra-tropics is still present and is also visible over most of the continents, thus confirming the conclusions from the comparison with the Karydis et al. (2011, 2017) data collection shown in Fig. 3.

The Fiedler et al. (2019) compared CDNC simulated by four global aerosol models with the Bennartz and Rausch (2017) dataset
and found biases of similar magnitude as for EMAC-MADE3, albeit with different geographical patterns which are specific to each model. The same was done by Kirkevåg et al. (2018) for the CAM5.3-Oslo model with the OsloAero5.3 aerosol scheme, who found a very similar pattern of biases in CDNC as in EMAC-MADE3, namely an overestimated (underestimated) CDNC over the tropical (extra-tropical) oceans. These authors also suggested that the lack of a satellite simulator in the model might affect the reliability of this comparison. Mulcahy et al. (2018) evaluated CDNC in the HadGEM3 model with the same two
MODIS-based satellite datasets used here and also found a better agreement with the Grosvenor and Wood (2018) retrieval than with the Bennartz and Rausch (2017) one, although their model performance over the continents appears better than for EMAC-MADE3. Finally, Zhao et al. (2018) compared the GFDL-AM4.0 model with Bennartz and Rausch (2017) and reported significantly underestimated CDNC, especially along the coastlines in the outflow of large emissions sources. In conclusion, the EMAC-MADE3 model's ability in simulating CDNC appears to be in line with the performance of other global models
reported in the literature, although the relative limited amount of data available for evaluating this quantity and the large uncertainties still affecting the satellite retrievals warrant further investigations in the future.

The current version of EMAC-MADE3 also allows to calculate the supersaturation in liquid clouds based on the $\kappa$-Köhler theory (Petters and Kreidenweis, 2007) as an alternative to the fitting function by Abdul-Razzak and Ghan (2000). An additional sensitivity experiment performed with this alternative formulation reveals no significant differences in the resulting
CDNC (see Figures S3 and S4 in the Supplement).

## 4.4   Ice cloud properties

Evaluating the microphysical properties of ice clouds by means of satellite data is a challenging task, due to the large uncertainties of satellite retrievals of such properties, mostly related to the difficulties in retrieving ice water content with passive instruments (Waliser et al., 2009). Given also that this model configuration has been especially developed for studies of

aerosol effects on cirrus clouds, to evaluate the model we use a collection of in situ measurements from 18 aircraft-based field campaigns compiled into a climatology by Krämer et al. (2009, 2016) and further complemented with more recent data (Krämer et al., 2020). A detailed description of the respective instruments is given in these publications. The campaigns took place in several locations including Europe, Australia, Africa, Seychelles, Brazil, USA, Costa Rica, and the tropical Pacific, i.e. in the latitude band between 75° N and 25° S, for a total of 113 flights. The measurements were performed in the cirrus regime between 182 and 243 K and include several cirrus properties such as ice water content (IWC, 127.5 h of measurements), number concentration of ice crystals (ICNC, 70.9 h), ice crystal radius ($R_{ice}$, 65.9 hours), as well as in-cloud and clear-sky relative humidity with respect to ice ($RH_{ice}$, 80.9 h and 157.8 h, respectively). Consistently with the measurements, in the model only the number concentration of ice crystals in the range 3–960 $\mu$m in terms of mean volume diameter is considered, where the mean volume diameter is defined in analogy to Eq. (6) of Lohmann et al. (2007).

The observational data are provided as probability distribution functions in bins of 1 K in the temperature range 182 to 243 K. Cloud variables in the model (IWC, ICNC, $R_{ice}$ and $RH_{ice}$) are sampled in the same range, considering only pressure levels with $p > 100$ hPa and selecting only the model grid boxes corresponding to the locations of the measurements used to generate the observational climatology. Following the same approach as K14, $RH_{ice}$ is calculated by the cloud parameterization from air pressure $p$, air temperature $T$, specific humidity $q$, and saturation specific humidity with respect to ice ($q_{ice}$) at each model time-step:

$$RH_{ice} = 100\frac{q}{q_{ice}}, \tag{22}$$

with

$$q_{ice} = \frac{0.622\, p_{ice}}{p - 0.378\, p_{ice}}, \tag{23}$$

where $p_{ice}$ is the temperature-dependent saturation vapor pressure over ice, calculated according to Murphy and Koop (2005). To distinguish between cloudy and cloud-free model grid boxes when comparing with the observations, the criterion IWC>0.5 mg kg$^{-1}$ is adopted.

The results of this comparison are shown in Fig. 5 for five variables. IWC simulated by the model is in remarkably good agreement with the observations across the whole temperature range reported in the data. The observations are, however, characterized by a larger spread in the distribution of the IWC values in each temperature bin: this is not surprising, since the model cannot capture the small-scale variability due to its coarse resolution. The median value is also in very good agreement with the observations, with a normalized mean bias (NMB[1]) of $-21.4\%$. A good agreement is also found for ICNC, at temperatures below 225 K, while for higher temperatures significant deviations are present, and the NMB of the medians is 177%. A consistent bias is found for the mean-volume radius of the ice crystals (NMB = $-40\%$), which is lower than in the measurements especially at higher temperature. This is due to the higher number of ice crystals in the simulations at comparable IWC. Relative humidity with respect to ice is very well captured by the model, both in the cloudy (NMB = 6%) and cloud-free

---

[1]The normalized mean bias is calculated as NMB = $100 \sum_i (M_i - O_i)/\sum_i O_i$, where $M_i$ and $O_i$ are the model and observation medians in each temperature bin, respectively.

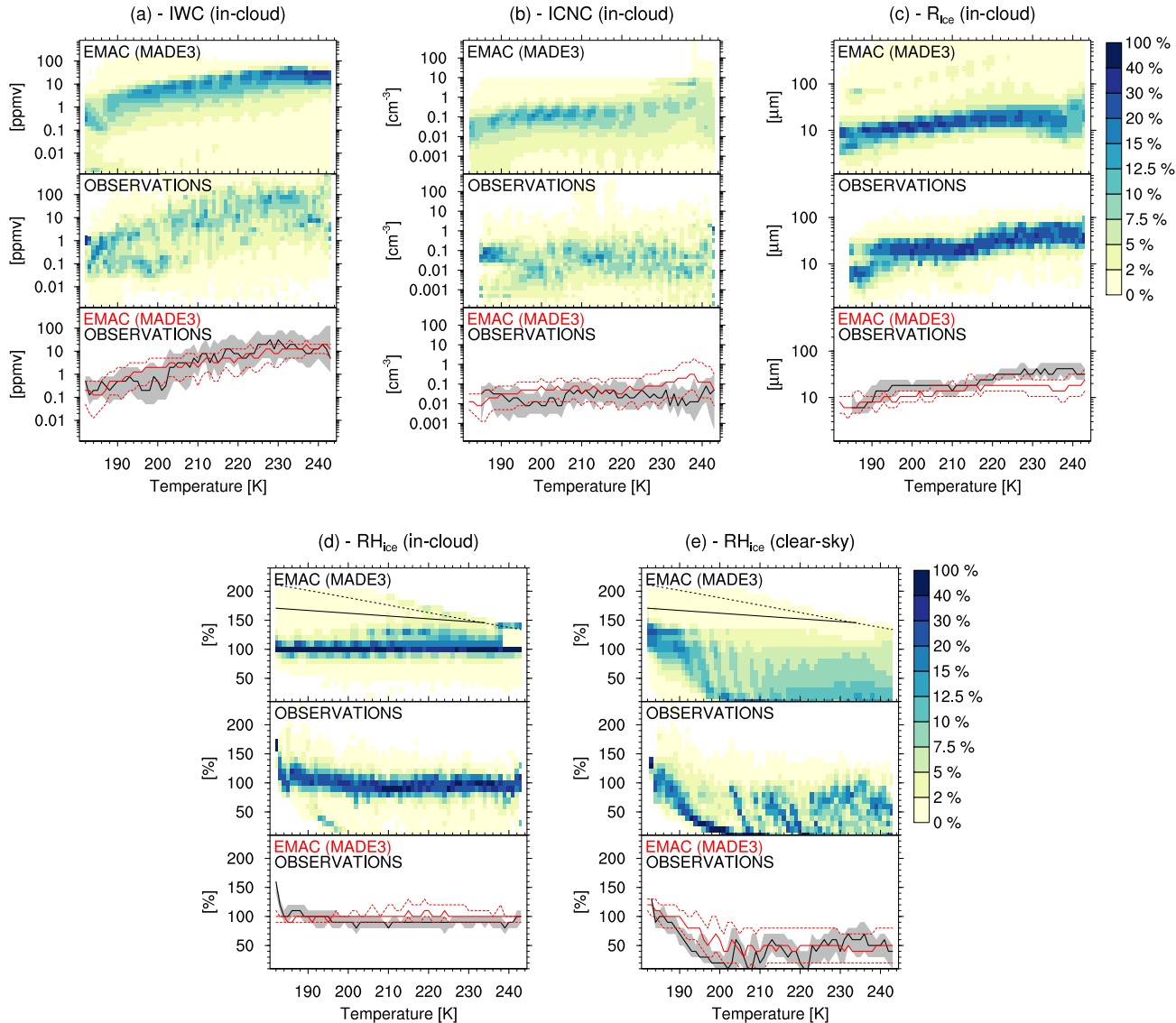

**Figure 5.** Climatology of various cirrus properties derived from flight campaigns as compared with EMAC (MADE3) simulations: in-cloud ice water content (a), in-cloud ice crystal number concentration (b), ice crystal mean-volume radius (c), in-cloud (d) and clear-sky (e) relative humidity with respect to ice. The data are plotted as probability distribution functions in 1-K temperature bins in the model (top plot in each panel) and in the observations (middle plot). The bottom plots in each panel show the median (solid red line) and the 25/75% quantiles (dashed red) of the model compared with the median (black solid line) and the 25/75% quantiles (gray shading) of the observations. The dashed and solid black lines on the relative humidity panels (d, e) represent the water saturation and the homogeneous freezing threshold (Koop et al., 2000), respectively.

(NMB = 18%) areas: however this is a feature which is largely controlled by the model dynamics (temperature and pressure) according to Murphy and Koop (2005, see Eqs.(22)-(23) above) and is therefore only indirectly related to the aerosol-cloud coupling which is evaluated in this study.

These findings are further supported by a similar comparison to recent in situ cirrus measurements performed during the cirrus campaign in mid-latitudes ML-CIRRUS (Voigt et al., 2017). Data from this campaign are also included in the cirrus climatology shown in Fig. 5 and discussed above (see also Krämer et al., 2016), but stem from a different set of instruments. The ML-CIRRUS climatology is based on more than 18 h of in situ cloud observations from 13 research flights in mid-latitude cirrus clouds over Europe and the Northern Atlantic. The ice crystal number concentration is determined from three cloud probes mounted on the wings of the aircraft: the Cloud and Aerosol Spectrometer with detector for polarization CAS-DPOL (Kleine et al., 2018), the Cloud Imaging Probe CIP and the Precipitation Imaging Probe PIP (de Reus et al., 2009; Weigel et al., 2016). These three instruments cover the ice crystal size range between 3 and 6400 $\mu$m. IWC is calculated from enhancement corrected total water measurements with the Water Vapor Analyzer (Voigt et al., 2014; Afchine et al., 2018) gas phase observations as described by Kaufmann et al. (2018). Relative humidity with respect to ice is determined from gas phase water vapor measurements with the Airborne Mass Spectrometer AIMS (Kaufmann et al., 2016). As for the global climatology, the ML-CIRRUS data for IWC, ICNC and RH$_{ice}$ are processed in 1 K bins, but for temperatures between 203 and 243 K, as observed in mid-latitudes. To account for different spatial resolutions of the cloud particle probes, measured ICNC are averaged with a running mean of 5 seconds. The model output is processed and compared using the same method as for the global climatology, but considering only spring months (March to May) over the simulation period. The results of this comparison are shown in Fig. 6: the agreement of simulated IWC and RH$_{ice}$ with ML-CIRRUS data is very good (NMB = −17% and 19%, respectively), supporting the results from the global climatology, although the model shows a negative (positive) bias at temperatures below (above) 225 K for IWC. Also the high bias in modeled ICNC at temperatures above 225 K is confirmed for the meteorological conditions in mid-latitude cirrus, although it is slightly lower (NMB = 139%) than in the global climatology. We further note that the ICNC measured by ML-CIRRUS is about a factor of two higher than in the global climatology: in the temperature range 203–243 K, the average of ICNC median values in ML-CIRRUS is 0.07 cm$^{-3}$, while is 0.03 cm$^{-3}$ in the global climatology. This difference is also found in the model simulations, albeit with higher values due to the aforementioned bias (0.16 versus 0.09 cm$^{-3}$). This is an interesting difference, which could be due to the specific meteorological and dynamic conditions encountered during the ML-CIRRUS campaign with respect to the global climatology, and their seasonality, but might also be a signature of an aircraft-induced increase in ICNC above continental Europe and in the Northern Atlantic flight corridor (see also Urbanek et al., 2018). This will be further investigated in a follow-up study on aviation impacts on cirrus clouds.

An evaluation of model simulations against the same data from the Krämer et al. (2009, 2016) climatology (see Fig. 5) was also performed by Bacer et al. (2018) for two different cirrus parameterizations implemented in the EMAC-GMXe global model. For temperatures above $\sim$ 225 K, the ICNC simulated by these two parameterizations is characterized by a high bias of about the same magnitude as the one found here. Biases are also found at lower temperatures, but they depend on the chosen parameterization. The same aircraft data was also used by Penner et al. (2018) to evaluate ICNC in the NCAR-CAM5.3, who

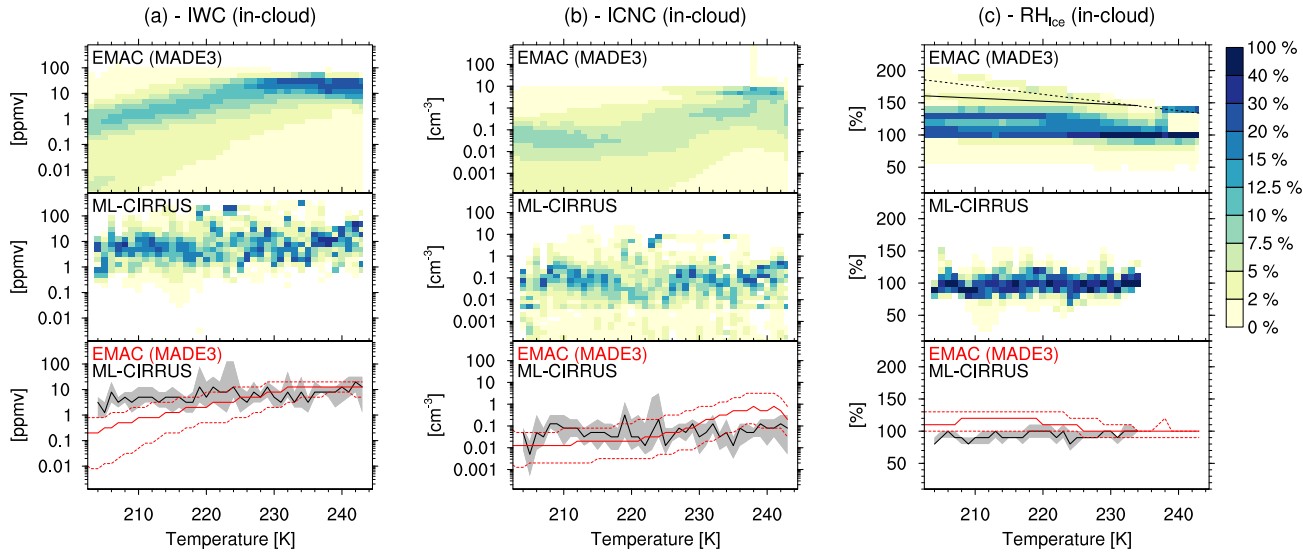

**Figure 6.** As in Fig. 5, but comparing to observational data from the ML-CIRRUS campaign. Shown are in-cloud ice water content (a), in-cloud ice crystal number concentration (b), in-cloud relative humidity with respect to ice (c).

found a significantly high bias around 200 K, relatively independent of the assumptions on the INP properties in their model. A previous version of the Krämer et al. climatology was used by K14 to evaluate the ECHAM5-HAM model with the same cloud scheme implemented here: they also reported a bias in ICNC at temperatures $T \gtrsim 220$ K, but underestimating ICNC compared to the observations rather than overestimating it as EMAC-MADE3, and ascribed it to underestimated homogeneous

5 freezing rates in the model, which in turn could be due to misrepresentation of temperature and/or vertical velocity. Although the two models share the same dynamical core (ECHAM5), differences in the dynamics could still arise due to the nudging mode adopted for the simulations in this study, as K14 performed their experiments in free running mode.

### 4.5 Precipitation

The pattern of precipitation (Fig. 7) is reproduced remarkably well by the model compared to the Global Precipitation Cli-

10 matology Project - Satellite and Gauge data (GPCP-SG; Adler et al., 2018), based on GPI, OPI, SSM/I and TOVS retrievals. Precipitation in EMAC is, however, characterized by a high bias in the tropics, especially over the Pacific and Indian oceans, and small negative bias in the extra tropics; this is consistent with the biases found for liquid water path in Sect. 4.2, which anti-correlate with the precipitation biases, as expected. Interestingly, a very similar pattern of biases was found by Lamarque et al. (2013) for the models participating in ACCMIP (Atmospheric Chemistry Climate Model Intercomparison Project).

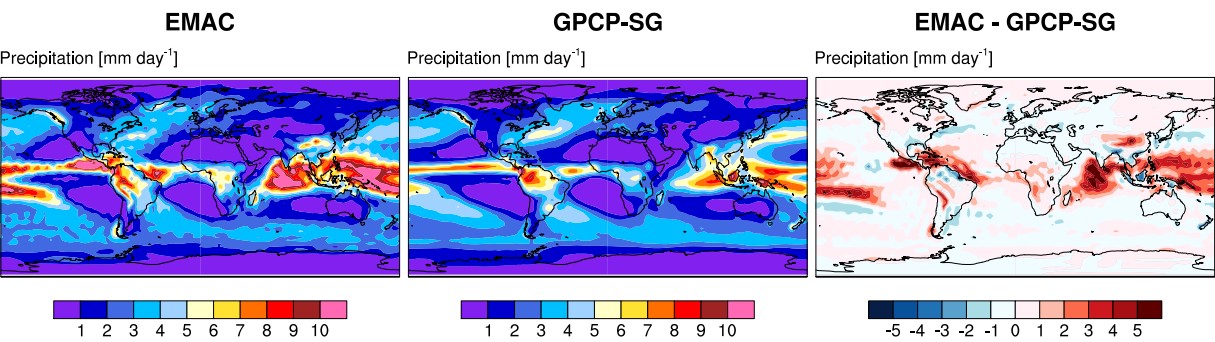

**Figure 7.** Similar to Fig. 2 but for precipitation.

## 4.6   Radiation

Figure 8 shows a similar comparison for radiation variables, namely shortwave (first row) and longwave (second row) cloud radiative effect compared to the CERES-EBAF satellite data based on MODIS, SNPP and NOAA-20 retrievals (Loeb et al., 2018). In the model, these quantities are given by the difference between the top-of-the-atmosphere all-sky and clear-sky fluxes, the latter being calculated via a second call to the radiation module which ignores the cloud effects. Although the general pattern is well captured by EMAC, the shortwave cloud radiative effect in the model is mostly weaker over the tropics and stronger at mid-latitudes than the observations, a picture which is consistent with the aforementioned ECHAM5 bias in low cloud cover (Räisänen and Järvinen, 2010). As noted by Räisänen and Järvinen, this is relatively independent of the cloud fraction scheme and can be ascribed to other model components, such as the convection scheme and the boundary-layer scheme (see also Righi et al., 2015a). It is therefore not related to the specific model configuration which is being evaluated in this work. Longwave cloud radiative effect is also reasonably well represented in the model, although with a generally positive bias, which is strong over Central America.

## 5   Anthropogenic aerosol ERF

As a further characterization of model performance, we calculate the anthropogenic aerosol ERF using the new model configuration. We quantify this as the difference in the top-of-the-atmosphere all-sky shortwave and longwave fluxes between the reference simulation and a similar experiment, where the 1850 (pre-industrial) emissions for anthropogenic and biomass burning sources are used instead of the 2000 (present day) ones. Other emissions, such as those from natural sources, are left unchanged between the two experiments. As mentioned in Sect. 2, radiatively active gases are also kept constant, so that the resulting ERF is solely due to changes in the concentrations of aerosols and the resulting cloud modifications.

This estimate results in an anthropogenic aerosol ERF of about $-1.42 \pm 0.03$ W m$^{-2}$. Considering only the cloudy-sky fluxes (i.e., diagnosing the change in the net cloud radiative effect), we obtain an ERF of $-0.96 \pm 0.02$ W m$^{-2}$. A comparison with the estimates of the IPCC AR5 (Boucher et al., 2013; Myhre et al., 2013) shows that EMAC-MADE3 simulates a more

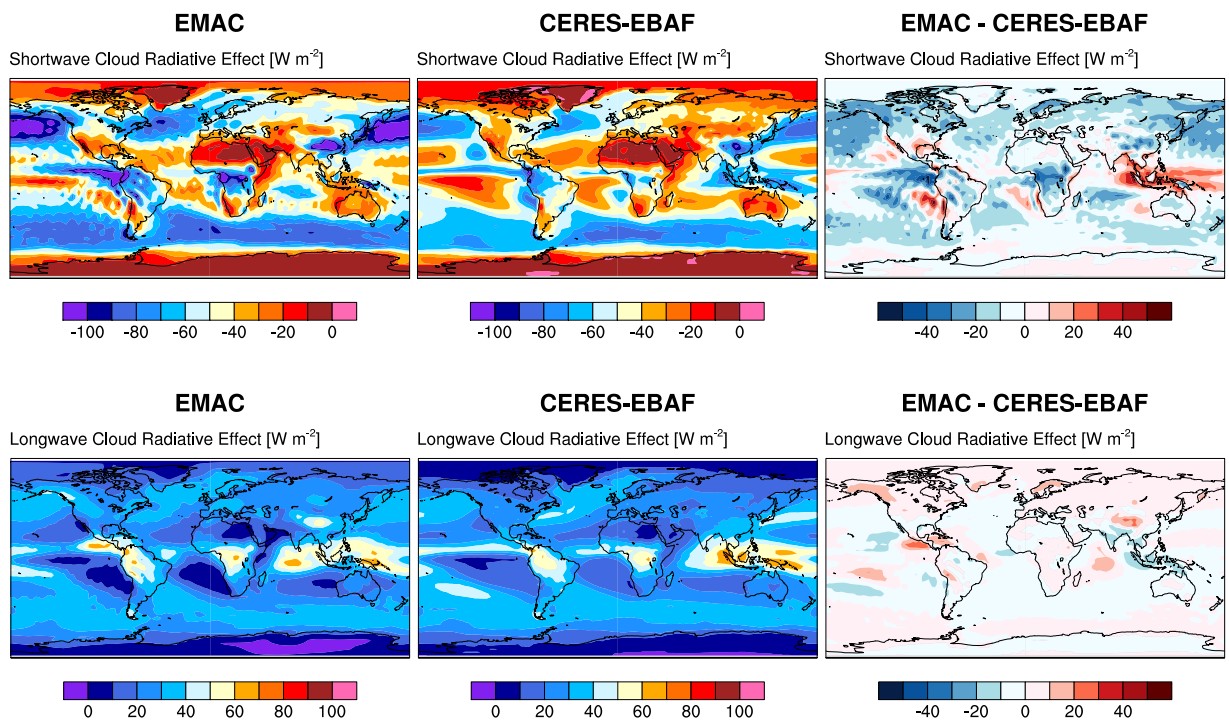

**Figure 8.** Similar to Fig. 2 but for radiation variables: shortwave (top) and longwave cloud radiative effects (bottom). Satellite data are averaged over the 2003-2015 time period.

negative aerosol effect: the aerosol ERF (sum of the effects from aerosol-radiation and aerosol-cloud interactions) reported by the IPCC and based on expert judgement is $-0.9$ W m$^{-2}$ with a 5 to 95% uncertainty range of $-1.9$ to $-0.1$ W m$^{-2}$. The effect due to aerosol-cloud interactions only amounts to $-0.45$ W m$^{-2}$, with a 5 to 95% uncertainty range from $-1.20$ to $0$ W m$^{-2}$. The recent review by Bellouin et al. (2019) considers additional observational constrains on the aerosol ERF and

5    reports a range of $-1.60$ to $-0.65$ W m$^{-2}$ ($-2.0$ to $-0.4$ W m$^{-2}$) for a 68% (90%) confidence interval, respectively, which basically confirms the IPCC ranges.

     This result shows that EMAC-MADE3 tends to simulate a comparatively high aerosol-induced cooling and so it could be more sensitive to changes in aerosol concentrations than other global aerosol-climate models. The high bias in liquid water path discussed in Sect. 4.2 and shown in Fig. 2 could be responsible for this high sensitivity, due to an overestimated cloud lifetime

10    effect. This is a common problem in coarse-resolution global models, which are not able to resolve the enhanced entrainment of dry air in clouds with higher CDNC, which would lead to droplet evaporation and thus partly offset the CDNC-induced increase in cloud lifetime (Ackerman et al., 2004; Xue and Feingold, 2006; Jiang et al., 2006; Altaratz et al., 2008; Mülmenstädt et al., 2019). Other studies show that the choice of the tuning parameters can also determine the resulting aerosol ERF: Hoose et al. (2009) and Neubauer et al. (2019) found a smaller (i.e., less negative) aerosol ERF when increasing CDNC$_{\mathrm{min}}$ from 10 cm$^{-3}$

$(-1.7\,\mathrm{W\,m^{-2}})$ to $40\,\mathrm{cm^{-3}}$ $(-1.0\,\mathrm{W\,m^{-2}})$ in the ECHAM6 model, while Golaz et al. (2011) reported a linear dependency of the aerosol radiative flux perturbation on the autoconversion threshold radius in the GFDL AM3 model.

The relatively large aerosol ERF simulated by EMAC-MADE3 needs to be kept in mind when applying the model to calculate the climate impacts of specific emission sectors, as it is planned. Another aspect, which will also be covered by upcoming application studies, is the role of cirrus clouds in the estimates of the climate impact of anthropogenic aerosol. As mentioned in the introduction, most of the CMIP5 models do not include aerosol interactions with ice clouds as EMAC-MADE3 is now able to do, which could also explain the larger sensitivity of our model to aerosol perturbations. The quantification of the cirrus effect under different assumptions on the ice nucleating properties of BC from various sources and other relevant ice nuclei is intended to be part of a follow-up study.

# 6 Conclusions

In this paper, a new version of the EMAC-MADE3 global aerosol model coupled with a new microphysical cloud scheme has been presented and evaluated. The new cloud scheme features a detailed parameterization for aerosol-driven ice formation within cirrus clouds. The new configuration has been tuned by varying three cloud microphysical parameters (rate of rain formation by autoconversion, of snow formation by aggregation, and the minimum allowed cloud droplet number concentration) and one aerosol parameter (the size of newly formed aerosol particles by the nucleation process). The optimal values for these four parameters have been chosen by analysing the normalized root mean square error between simulated and observed key cloud and radiation variables. The evaluation of these model variables was performed using a comprehensive set of observations from satellite retrievals and in situ measurements, including data from aircraft-based field campaigns.

The main conclusions on the performance of the coupled version of EMAC-MADE3 can be summarized as follows:

1. EMAC-MADE3 is able to reproduce the global pattern of the main cloud and radiation variables in comparison with satellite and in situ data;

2. specific deviations, in particular in the representation of liquid water path which could point to an overestimated cloud lifetime, mostly confirm known biases of the ECHAM5 model and can therefore not be attributed to the new cloud scheme introduced in this work;

3. a more detailed evaluation of cloud variables in the cirrus regime against an aircraft-based climatology of in situ measurements demonstrates the ability of EMAC-MADE3 to adequately represent ice water content and ice crystal number concentration in cirrus clouds over a wide range of temperatures, albeit with a positive bias for the ice crystal number at higher temperatures;

4. the overall performance of EMAC-MADE3 in simulating global cloud and radiation variables is in line with the results of the CMIP5 models;

5. model biases in the representation of cirrus clouds are common to other models, such has ECHAM5-HAM, EMAC-GMXe, and NCAR-CAM3.5, using various parameterizations for aerosol-induced ice formation in cirrus clouds.

As a first application of the new model system, the anthropogenic aerosol effective radiative forcing has been calculated and found to be within the range reported by the IPCC AR5 based on an expert judgment, demonstrating the capabilities of the model to adequately simulate aerosol-induced impacts on the climate system. More targeted applications will include the simulation of the impact of individual emission sectors, such as aviation, on aerosol and clouds, with a specific focus on cirrus clouds, and are intended to be the subject of follow-up studies.

Despite the encouraging performance of this new model configuration, several uncertainties remain and have to be addressed in the future by targeted applications of EMAC-MADE3. This includes, for instance, the tendency of the model to simulate a relatively large anthropogenic aerosol radiative forcing, which is a common feature in coarse-resolution global models and may affect the estimates of the climate impacts of specific sectors, such as aviation. The model biases in CDNC and ICNC could also influence the model's ability to reproduce observed cloud radiative properties. This deficiency needs to be further reduced in future studies. Another limitation is related to the uncertain properties of ice nucleating particles, which in the case of black carbon still lack a sufficient level of understanding. Furthermore, the cirrus parameterization implemented here is based on a supersaturation threshold for ice nucleation, but it is not able to follow the ice formation process in detail by means of, e.g., a nucleation spectrum. This means that a single critical value is provided for each ice nucleating particle type, but this does not fully represent the complexity of the actual physical process. Finally, in this study we focused on ice formation in cirrus clouds from the perspective of aerosol particles driving this process, but it should not be forgotten that mesoscale and large-scale atmospheric dynamics play an equally (or even more) important role in the microphysics of cirrus clouds.

## 7  Code availability

MESSy is continuously developed and applied by a consortium of institutions. The usage of MESSy, including MADE3, and access to the source code is licensed to all affiliates of institutions which are members of the MESSy Consortium. Institutions can become members of the MESSy Consortium by signing the MESSy Memorandum of Understanding. More information can be found on the MESSy Consortium Web-site (http://www.messy-interface.org). The model configuration discussed in this paper has been developed based on version 2.54 and will be part of the next EMAC release (version 2.55).

The Earth System Model eValuation Tool (ESMValTool) v1.1.0, used to produce Figures 2, 4, 7, and 8 is available at https://github.com/ESMValGroup/ESMValTool/releases/tag/v1.1.0.

## 8  Data availability

The model simulation data analyzed in this work are available at https://doi.org/10.5281/zenodo.3630106.

*Author contributions.* MR conceived the study, implemented the new cloud scheme in EMAC, designed and performed the simulations, analyzed the data, evaluated and interpreted the results, and wrote the paper. JH conceived the study, contributed to the interpretation of the results and to the text. UL provided the new cloud scheme, contributed to design the simulations and to interpret the results; CB, with the help of BH and IT, developed and implemented the method for filtering online dust emissions at low model resolutions in EMAC; MK and CR provided the in situ data for the evaluation of the cirrus properties and contributed to the interpretation of the results; MP contributed to the design and the interpretation of the tuning experiments; VH, RH and CV provided the DACCIWA and ML-CIRRUS data for the evaluation of cloud droplet number concentrations and of cirrus properties, and contributed to the interpretation of the results and to the text.

*Competing interests.* The authors declare that they have no conflict of interest.

*Acknowledgements.* This study was supported by the DLR transport programme (projects *Transport and the Environment - VEU2* and *Transport and Climate - TraK*), by the DLR space research programme (project *Climate relevant trace gases, aerosols and clouds - KliSAW*) by the DLR aviation research programme (project *Eco-efficient air travel - Eco2Fly*) and by the Initiative and Networking Fund of the Helmholtz Association through the project *Advanced Earth System Modelling Capacity (ESM)*. The EMAC simulations were preformed at the German Climate Computing Center (DKRZ, Hamburg, Germany). The ML-CIRRUS campaign was supported by DFG SPP HALO1294 contract no VO1504/4-1, and Romy Heller by the EU ICE-GENESIS project within H2020 grant agreement no. 824310. The contribution of cloud sonde data from ML-CIRRUS by Stephan Borrmann and Ralf Weigel (University of Mainz and MPI-C, Germany) is kindly acknowledged. We are grateful to Yvonne Boose (DLR, Germany) for her comments and suggestions on an earlier version of the manuscript, and to Klaus Gierens, Patrick Jöckel, Axel Lauer, Matthias Nützel (DLR, Germany), Holger Tost (University of Mainz, Germany), Sara Bacer (MPI-C, Germany) and Miriam Kuebbeler for helpful discussions. The development work presented in this paper has greatly benefited from the support of the whole MESSy team of developers and maintainers.

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
