# Peer review of "Coupling aerosols to (cirrus) clouds in the global aerosol-climate model EMAC-MADE3"

_Geoscientific Model Development, 2019_

## Referee Comment (RC1) · Anonymous Referee #1 · 14 Sep 2019

Righi et al. implemented a parametrization of deposition nucleation into the EMAC model, although there seems to be insufficient support for such a parametrization based on observations. The new model fails to reproduce the observed dependence of ice effective radii on temperature and it is not clear what effect the assumption that soot acts as INP has on the results. Additional runs without the assumption that soot is an efficient INP and major clarifications are needed before I can recommend this manuscript for publication.

Major comments:

1) Page 3, lines 4 to 9: What was the rationale behind implementing the Hendricks et al. (2011) parameterization in EMAC-MADE and applying it to soot? Evidence from observations suggests that the paramterized process is not effective. I would expect

this model to compute an unrealistically high estimate of the impact of black carbon emissions from aviation on climate. Or am I missing something? Is there perhaps something that makes you think that the ice nucleation rates that one might expect from pore condensation are similar to the ones one might expect from condensation nucleation? Parameterizing pore condensation requires information on porosity and pore condensation is likely to be important for porous particles such as dust (David et al., 2019). Please explain this better.

2) The aerosol effective radiative forcing (ERF) computed with EMAC-MADE3 is $-1.76 \pm 0.04$ Wm$-2$. How big is the contribution of assuming that soot acts as INP? Please also state the ERF of soot emissions from aviation with and without assuming that soot acts as INP. The impact of assuming soot to be an efficient INP in EMAC-MADE must become clear somehow, and I don't think that asking for these very specific results from these sets of additional sensitivity runs (which are easily set up) is asking too much. In my opinion it is very important here to quantify the effects of these specific model development choices on the key model results.

3) In the light of my first major comment, I strongly suggest to use a model version without the assumption of soot being an efficient INP as the base version, and to present the simulation with soot INP as a sensitivity study.

4) Fig 2: the very pronounced north-south asymmetry of LWP is inconsistent with observations and with the bulk of CMIP5 models. This indicates that the cloud lifetime effect is probably too strong in EMAC-MADE3. Please discuss.

5) On the one hand, I appreciate that you chose a lower CDNCmin than ECHAM-HAM (Neubauer et al., 2019, doi:10.5194/gmd-12-3609-2019, see also https://doi.org/10.5194/gmd-2018-307-RC1 on this point). On the other hand, I wonder how you can justify a radiative imbalance as large as 5 W/m^2. Please explain.

6) Fig 5: Observations suggest a clear dependence of R_ice on temperature (e.g. lower panel in Fig. 5c), which I have also seen reproduced in a global model with a

two-moment ice microphysics (Salzmann et al., 2010, doi:10.5194/acp-10-8037-2010). EMAC-MADE3, on the other hand, does not reproduce this dependence. Instead, EMAC-MADE3 slightly overestimates R_ice at very low temperatures and strongly underestimates R_ice at higher temperatures. Both seems consistent with various heterogeneous nucleation processes being too efficient. (Overly efficient heterogeneous nucleation at very low temperatures could in principle suppress homogeneous nucleation, which can lead to an underestimate of the ice numbers.). Too small ice crystals in the mixed phase regime may also affect LW-ERF estimates. Please discuss.

Other comments: Perhaps move p. 3, l 26 ff to p3, l7? On the other hand, it would be better to re-write this entire paragraph taking into account the major comments above.

p. 19, line 9: $-1.76 \pm 0.04$ Wm$-2$: please discuss this result in the light of major points 2,3, and 4.

Fig 2: it would be good to also show zonal mean LWP plots.

p. 1, lines 9 to 11 ("The performance ..."): please refer to my major comment #4 regarding the hemispheric asymmetry of the simulated LWP above.

p. 1, line 10f: please be specific

p. 1, lines 12ff: please refer to my major comments #4 and #5 above and to my comment regarding p. 20, line 13.

p. 7, lines 14ff: "Since the EMAC-MADE3 ....": this seems pure speculation. A least one would have to show that these biases do not occur in models that use different emission data sets in a comparable AMIP-style setup.

Fig 3: since the model values are mean values, I suggest to include vertical bars which give some measure of the spread between the model values (e.g. standard deviations or quantiles).

p. 10, l. 6: as an aside: rather than only matching observations it would perhaps be

better to include a process-oriented approach in the future. For example, if the warm rain formation rate is overestimated compared to observations and too little rain forms via the ice phase, this indicates that the cloud lifetime effect may be overestimated. On the other hand, tuning strategies taking such process-oriented metrics into account are only starting to emerge.

p. 17, l. 2: would it be feasible to regrid the observations to model resolution in order to avoid this problem?

p. 17, l. 12: the simulated RH is also controlled by the large-scale condensation scheme, the assumption regarding cloud phase in the saturation adjustment, and other microphysical terms. In the original ECHAM5, the assumption regarding cloud phase in the saturation adjustment is based on a threshold for the mass mixing ratio of cloud ice, but this is different in EMAC. Line 18 on page 15 says that EMAC allows super-saturation over ice based on Murphy and Koop (2005) and supersaturation over ice is also seen i Fig. 6c. The simulated supersaturation depends on this parameterization.

p. 18: line 15: are you comparing a model with prescribed SSTs to the CMIP5 coupled models or is precipitation reproduced remarkably well also when compared to AMIP-style simulations with other models?

p. 18, l. 21: see my comment regarding p. 7, lines 14ff

p. 19, l. 6: I tend to think of the aerosol radiative forcing as the direct forcing. I think what you computed is better called effective radiative forcing (ERF), which by definition includes fast adjustments.

p. 19, l. 5: I assume you are using the standard double calls to estimate cloud radiative effects. Or does Dietmüller et al. (2016) do something special? If yes, please explain. If no, please mention that this is a standard method, so that people who know this method won't have to check Dietmüller et al. for details.

p. 20, line 13: "the total aerosol RF reported by the IPCC ranges between $-2.05$ and

0.05 Wm−2". Please be specific. How does this fit with AR5 Table 7.4? Also note that in the model with the largest ERFari+aci in AR5 Table 7.4, the ocean heats the atmosphere in the 20th century (Fig. 4, Golaz et al., 2013, doi.org/10.1002/grl.50232) in a setup that uses basically the same aerosol activation scheme, but a different microphysics scheme.

p. 21, l. 3: tuned for what? Usually, I would think about TOA balance. But the model is very much out of balance. I think you need to make clear that by tuned you mean tuned mainly to match LWP, IWP, ICNC, SWCRE, and LWCRE to values that were derived from satellite (remote sensing) observations. The large TOA imbalance of 5.23 W/mˆ2 precludes a successful coupling to an ocean component without further (fairly drastic) tuning measures.

p. 21, l. 7: satellite measurements: space-borne instruments measure radiation. Not even clear-sky radiation in a cloudy scene (as used to compute CREs in the model) is among the measured quantities.

p. 21, list item 1: please see my major comment #4 above.

p. 21, list item 3: please see my major comment #6 above. (I realize that a part of this point is implicitly included in the last sentence. But I still think the unrealistically weak dependence of effective radius on temperature is notable. )

p. 21, list item 4: here you are explicitly referring to "coupled" models from CMIP5. In the context of CMIP5, "coupled" usually refers to the fact that the models are run with an ocean component. Because EMAC-MADE3 is run in atmosphere-only (AMIP) mode (with fixed SSTs), cloud biases (e.g. the double ITCZ) should definitely be smaller than in coupled models. Note, however that in addition to results from coupled model experiments, the CMIP5 archive also includes results from dedicated atmosphere-only (AMIP) model experiments. In order to justify the statement that you are making here, you should compare EMAC-MADE either to uncoupled CMIP5 models (AMIP runs) or else to uncoupled (i.e. without ocean component) AeroCom experiments.

p. 21, list item 5: please be specific

p. 21, item 6: where is this shown?

p. 21, line 4 from bottom: please see my comment regarding p. 20, line 13.

p. 21, line 2 from bottom: please see my major comment #1.

p 23ff: it would be nice to analyze how much each of the processes contributes to the ice crystal number at a given temperature. One could for example plot time averaged process rates on the x-axis and temperature on the y-axis.

The conclusion section should include a discussion of uncertainties which are specific to this new model development effort. Furthermore, there are some general caveats to this type of study that could be mentioned in the introduction section. In particular, numerous studies suggest that in cases in which the cloud lifetime is limited by turbulent mixing and not by precipitation, coarse scale models may overestimate the cloud lifetime effect (see e.g. references in Salzmann et al., 2010, doi:10.5194/acp-10-8037-2010). Other studies have shown that the aci component of ERFari+aci in one model depends on the CDNCmin tuning parameter (Neubauer et al., 2019, doi:10.5194/gmd-12-3609-2019, Hoose et al., 2009, doi:10.1029/2009gl038568) and in another model on the autoconversion threshold (Golaz et al., 2010, doi: 10.1175/2010JCLI3945.1).

Technical: p. 8, l. 19: calculate -> calculated Correct line numbers would have helped with the review.

---

## Referee Comment (RC2) · Anonymous Referee #2 · 2 Oct 2019

Review of "A new approach to simulate aerosol effects on cirrus clouds in EMAC v2.54" by Mattia Righi et al.

This manuscript presents a new version of a Chemistry Climate Model, focused on developments of the ice cloud scheme, and integrates previous work. It is a not to insightful model description paper, that doesn't really describe any thing in detail. The analysis methods are not state of the art, and somewhat inconsistent. The paper needs major revisions if it is to be suitable for GMD.

General concerns:

1. The paper is all about clouds and aerosols, but I cannot find how activated aerosols are connected to cloud and crystal number. Please state that explicitly.

[Figure]

2. The information in Appendix A is only referred to in passing, and should be a larger part of the manuscript

3. The sensitivity tests and tuning exercises are laudable, but the description of optimizing the model and what parameters are chosen is ad-hoc and subjective. It doesn't seem like any formulation matched the data, but no rigorous method was used to get the final tuned model.

4. Some of the plots show significant artifacts (oscillations) in averaged model fields. I am hoping that this is just a product of the 11 hour time sampling, but it means there is either something fundamentally wrong with either the model itself or the analysis methods.

5. Since this is about ice clouds and aerosols, please determine the ice cloud ERF from anthropogenic aerosols. Techniques exist for this.

6. Substantive comments below on particular aspects of the data analysis also need to be considered in a revision.

Specific Comments:

Page 3, L16: this does not go back far enough. Liu et al (2005) developed a parameterization of homogeneous and heterogeneous freezing that was implemented in CAM by Gettelman et al (2010).

Liu, X., and J. E. Penner. "Ice Nucleation Parameterization for Global Models." Meteor. Z. 14, no. 499–514 (2005).

Page 3, L34: The abstract mentions the mixed phase, but there is no discussion here. This seems to be all cirrus clouds.

Page 6, L33: please state how the aerosols and clouds are connected. You detail how INP are derived. But once you have INP how are they linked with the cloud scheme? Do you just set the cloud drop and crystal number to be at least as large as the activated

number?

Page 7, L14: since the INP parameters are the new part of this study, I think a bit more detail is warranted here, maybe a paragraph, and not left for the appendix.

Page 8, L4: so what are you really focusing on? Not clouds? Not aerosols? What do you mean by the 'interaction'. At this point as a reader I still have not idea. Please be more specific.

Page 8, L23: please at least mention what the data sources are (CERES, MODIS, etc)

Page 8, L24: so how many simulations were conducted? 20? How long are they? Why not use a more sophisticated method for sampling that can better sample a wider parameter space following Carslaw, Regare, etc.?

Page 9, Fig1: LWP needs a range of uncertainty.

Page 10, L5: so was there an objective way the parameters are chosen? It's not clear how you made a decision. Wouldn't a root mean square error be better than means? How do you know there is not cancellation?

Page 11, L4: Cloud fraction should only be compared if the model and the data are thresholded identically, usually requiring use of a satellite simulator.

Why is there a significant oscillation in mean cloud fraction and LWP fields over the subrropical oceans in the model? That needs to be explained.

Page 13, Figure 3: gray does not stand out from dark blue. I can not distinguish the DACCIWA point. Also, can you give some idea of regime or location, maybe another plot with difference from the 1:1 line as a function of latitude?

Page 14, L5: so the reason for the high SWCRE is likely the high CDNC? Is that a correct statement?

Page 14, L14: in figure 4, the gradients are wrong in EMAC over most of the oceans.
CDNC drop with latitude in EMAC but increase with latitude in the observations. What is going on? This seems like a fundamental error somewhere either in model or data that needs explanation.

Page 15, L5: I'm not sure figure 5 is remarkably good. There are some missing modes in the observations that EMAC doesn't have, and this is a qualitative comparison on a log scale. There are factors of 2 error here. Also, the model needs a range from variability like the observations, not just a line for the mean.

Page 18, L15: for Figure 7 and all the subsequent maps: can you calculate. Root mean square error for all the fields. This is useful for quantitatively understanding pattern biases.

Page 19, Fig 7: why is EMAC so noisy in its cloud fields? If it is sampling then you should use a different method. If it is not a sampling issue, then something is wrong with the model. I've seen this before with radiation code errors leading to jumps in solar zenith angle and incident radiation...

Page 19, L11: does this mean the cloud effect is -1,13 and he rest (-0.63) is A direct effect? That seems large.

Page 20, L23: however, Gettelman et al 2012 found that the ice cloud indirect effect in CAM and ECHAM-HAM is positive. Please document the ice cloud portion of the ACI if you can (a simple simulation where you fix the IN and one where it varies) and compare to previous work, since that seems to be the major development in this paper.

Gettelman, A., X. Liu, D. Barahona, U. Lohmann, and C. C. Chen. "Climate Impacts of Ice Nucleation." Journal of Geophysical Research 117, no. D20201 (2012). https://doi.org/10.1029/2012JD017950.

Page 21, L31: I'm not sure how the parameters were chosen. This should be more objective.

---

## Referee Comment (RC3) · Anonymous Referee #3 · 29 Oct 2019

This paper presents the implementation of a new cloud microphysical scheme for parameterizing ice formation in cirrus clouds for the use in the global chemistry model EMAC. The new scheme takes into account the aerosol type that serves as INP and the competition of homogeneous and heterogeneous ice formation that is relevant in cirrus clouds. As part of the model evaluation, results from a tuning exercise are reported, as well as the comparison to observational data.

The paper fits within the scope of GMD and tackles a challenging topic that is relevant for the community. I have the following comments that should be addressed before the paper can be accepted for publication:

1. As the authors state, the ice nucleating properties of BC are highly debated. In light of these uncertainties, it makes more sense to define the reference case without BC

as INP and show a sensitivity case where the impact of BC as INP is included. How important is BC as INP using the current assumptions?

2. The aerosol representation of MADE3 is quite complicated and an interesting feature of the work. I suggest explaining this in more detail (even though it has been explained in previous papers). In this context, the sentences on p. 23/lines 20-23 are hard to understand. Please clarify. It would also be interesting to learn more about if keeping track of the aerosol mixing state as done with MADE3 (as opposed to only tracking average composition using fewer modes) is yielding improved results.

3. Given that this is a model development paper on a new cloud microphysical scheme, more emphasis should be given presenting the underlying set of equations. Some of this is currently presented in Appendix A. This should move to the main manuscript, but more needs to be added to explain how the "potential ice-nucleating particles" are used to produce ice crystals. Overall the presentation of the modeling equations is fragmented and hard to follow.

4. More details need to be provided for the model tuning in section 4.1. What is the justification to choose specifically these four tuning parameters? And given that for each of the four tuning parameters five values are chosen, do you perform all 20 simulations (one for each parameter calculation)? Have you considered using a latin-hypercube sampling strategy of the parameter space of your tuning parameters? And on what basis is the optimal tuning configuration chosen?

5. How has including the new scheme changed the results compared to the previous version of the model? I.e. is the "new approach" an improvement over the old approach?

---

## Author Comment (AC1) · 30 Jan 2020

***A new approach to simulate aerosol effects on cirrus clouds in EMAC v2.54***
**(now: *Coupling aerosols to (cirrus) clouds in the global aerosol-climate model EMAC-MADE3*)**
**M. Righi et al.**
**Replies to referees' comments**

We are grateful to the three reviewers for their insightful comments and constructive criticism, which greatly helped us to improve the manuscript.

Before addressing each comment in detail below, we would like to summarize three major changes introduced in the revised version of the manuscript:

1.  We improved the model performance in reproducing cirrus properties when compared with in situ data: we performed many test simulations acting on the different processes controlling aerosol-induced ice formation in cirrus clouds and finally concluded that increasing the model vertical resolution significantly reduces the bias in ice crystal number concentrations. This also helped to better reproduce the observed dependence of ice crystal radii on temperature, as advocated by Reviewer #1. In the revised version we therefore discuss a configuration with 41 instead of 19 vertical levels, which is specifically designed to better resolve the upper troposphere. This change in the model configuration required the repetition of the tuning exercise (see next point), resulting in a slightly different choice of the tuning parameters for the reference simulation, and the re-evaluation of the model. The main conclusions on the model performance discussed in Sect. 4.1-4.3, however, do not change significantly, while considerable improvements are found concerning the representation of cirrus properties in EMAC-MADE3 (Sect. 4.4).
2.  Since the tuning exercise had to be repeated due to the new model resolution in use, we took the opportunity to completely revise our tuning approach, following the reviewers' suggestions about this. We are now using quantitative statistics (normalized RMSE and relative standard deviation) to identify the most important tuning parameters for each variable and to choose the optimal setup. We also discuss and quantify the impact of nudging on the model radiative balance in more detail.
3.  We significantly extended the model description part (Sect. 2), to meet the requests from all reviewers, who suggested providing more details about the cirrus parametrization and the representation of ice nucleating particles in the model. The details previously given in Appendix A are now integrated in the main text in Sect. 2.

Other less extensive but important changes are highlighted below while addressing the single reviewers' comments. Reviewers' comments are marked in blue, authors' reply in black, and text quotes in *"italic red"*.

**Anonymous referee #1**

*Righi et al. implemented a parametrization of deposition nucleation into the EMAC model, although there seems to be insufficient support for such a parametrization based on observations. The new model fails to reproduce the observed dependence of ice effective radii on temperature and it is not clear what effect the assumption that soot acts as INP has on the results. Additional runs without the assumption that soot is an efficient INP and major clarifications are needed before I can recommend this manuscript for publication.*
We performed several simulations varying the ice nucleating properties of soot, but this neither helped to reduce the bias in ice crystal number concentration (ICNC) nor to reproduce the dependence of ice crystal radii on temperature. This was achieved, however, by increasing to model vertical resolution to 41 vertical levels, thus better resolving the upper troposphere: this significantly reduces the normalized mean bias (NMB) in ICNC from 592% in the previous version to 177% in the current one (Fig. 6a), also better capturing the variation of radii with temperature (Fig. 6b).

*1) Page 3, lines 4 to 9: What was the rationale behind implementing the Hendricks et al. (2011) parameterization in EMAC-MADE and applying it to soot? Evidence from observations suggests that the parameterized process is not effective. I would expect this model to compute an unrealistically high estimate of the impact of black carbon emissions from aviation on climate. Or am I missing something? Is there perhaps something that makes you think that the ice nucleation rates that one might expect from pore condensation are similar to the ones one might expect from condensation nucleation? Parameterizing pore condensation requires information on porosity and pore condensation is likely to be important for porous particles such as dust (David et al., 2019). Please explain this better.*
The goal of this development effort was to make EMAC-MADE3 suitable for application studies on the impact of aviation soot on cirrus clouds, and more in general on the role of INPs for climate. Although the topic is still debated, there are several laboratory studies suggesting that anthropogenic soot (including aviation soot) may act as an INP in certain conditions. Uncertainties are large also on the modelling side and there is still no consensus on the actual impact of (aviation) soot on climate. In order to better motivate our study and to put it in the context of the existing literature on this topic, we extended the corresponding paragraphs in the introduction as follows: *"The ice-nucleating properties of the different types of BC are still highly debated, but several laboratory studies suggest that BC may act as INP at typical cirrus temperatures. Möhler et al. (2005a, b) investigated the ice nucleating properties of coated and uncoated soot in the AIDA cloud chamber and found that uncoated soot is able to nucleate ice at low ice saturation ratios between 1.1 and 1.3, but pointed out that coating with sulfuric acid and mixing with organic carbon increase the ice supersaturation threshold. This is supported by the measurements of Crawford et al. (2011), who also used the AIDA cloud chamber to study the ice nucleation of coated and uncoated propane flame soot and reported a 1% nucleated fraction of uncoated low organic carbon soot at saturation ratios as low as 1.22, while they measured lower ice formation efficiencies for soot with higher organic carbon content and for coated soot. Koehler et al. (2009) analysed different soot types and observed ice nucleation below the homogeneous freezing threshold for some of them (including TC1 soot resulting from burning of aviation kerosene). In an intercomparison study among different instruments, Kanji et al. (2011) reported ice nucleation on graphite soot at supersaturations between 1.3 and 1.5. Chou et al. (2013) considered fresh and aged diesel soot particles and measured ice nucleation fractions of several percent at ice saturation ratios around 1.4. Also Kulkarni et al. (2016), analysed the ice formation ability of diesel soot under cirrus conditions and reported a 1% frozen fraction of the soot particles at similar ice saturation ratios. Nichman et al. (2019) examined six types of BC particles considered as proxies for atmospheric BC and found onset saturation thresholds for ice nucleation between 1.1 and 1.5. Recent studies observed BC nucleation at cirrus temperatures, but explained it with pore condensation and freezing rather than with deposition nucleation (Wagner et al., 2016; Marcolli, 2017; Mahrt et al., 2018; David et al., 2019). As shown by Mahrt et al. (2019), the process of pore condensation and freezing will become more important after cloud processing of soot, which enhances soot-induced heterogeneous ice formation at cirrus temperatures by reducing the saturation threshold for ice nucleation.*

*Despite the uncertainties resulting from the large range of the measured ice formation abilities of BC in the cirrus regime, the laboratory studies clearly reveal that effects of soot on ice cloud formation are potentially relevant and the resulting climate impacts could be significant, especially when considering specific emission sources such as aviation (Koehler et al., 2009; Hendricks et al., 2011; Zhou and Penner, 2014; Penner et al., 2018; Urbanek et al., 2018) or land transport (Chou et al., 2013; Kulkarni et al., 2016).*

*There are, however, only few global models capable of simulating aerosol-induced ice formation in the cirrus regime in detail. Liu and Penner (2005) developed a parameterization for homogeneous and heterogeneous ice nucleation, which was later implemented in the CAM model by Gettelman et al. (2010). More recently, Bacer et al. (2018) coupled the Global Modal aerosol eXtension (GMXe) submodel (Pringle et al., 2010) to the cirrus parameterization by Barahona and Nenes (2009) in the EMAC model, opening interesting perspectives for comparing different aerosol and cloud microphysical schemes within the same model framework. Several studies attempted to quantify the climate impact resulting from the influence of BC on cirrus clouds, but these estimates are quite diverse and there is no consensus on the magnitude, and not even on the sign, of this, effect. Using the CAM3 model coupled with the IMPACT aerosol model, Liu et al. (2009) simulated the impact of soot on cirrus clouds and found a significant warming effect, strongly dependent on the assumptions on the ice nucleating ability of soot itself. Using offline calculations, Penner et al. (2009), however, argued that soot impact on cirrus clouds result in a significant cooling effect, while Hendricks et al. (2011, with the ECHAM4 model) and Gettelman and Chen (2013, with the CAM5 model) found no statistically significant climate effects. Zhou and Penner (2014) discussed the role of background INPs as an important source of uncertainty affecting the estimates of aviation soot impacts on climate. Penner et al. (2018) included an ice-formation parameterization for cirrus clouds in the NCAR-CAM5.3 model coupled to the IMPACT aerosol model, also distinguishing three aerosol mixing states in three size modes. Their resulting estimates of the radiative forcing from various emission sources show a large negative climate impact due to aerosol-induced cirrus modifications."*

In preparation for a follow-up study, we performed a first set of simulations with the new model configuration and found that the aviation impact is not unrealistically high as suggested by the reviewer, but it is rather close to the statistical significance limit and much lower than other existing model-based estimates, e.g. the recent Penner et al. (*J. Geophys. Res.*, 2018) study. This strongly depends, of course, on the assumptions on the ice nucleating properties of aviation soot, which are still highly uncertain and will be explored in detail in the follow-up study by means of sensitivity experiments.

2) The aerosol effective radiative forcing (ERF) computed with EMAC-MADE3 is −1.76±0.04 Wm−2. How big is the contribution of assuming that soot acts as INP? Please also state the ERF of soot emissions from aviation with and without assuming that soot acts as INP. The impact of assuming soot to be an efficient INP in EMAC-MADE must become clear somehow, and I don't think that asking for these very specific results from these sets of additional sensitivity runs (which are easily set up) is asking too much. In my opinion it is very important here to quantify the effects of these specific model development choices on the key model results.

We agree with the referee that estimating the ERF of soot aviation emissions would be an interesting result, but it is not within the scope of this publication. The goal of this paper, as stated in the introduction, is to document, tune and evaluate the new model configuration, assessing its strengths and weaknesses for future reference. This was also the reason for submitting it to GMD, which has exactly this scope. Additional sensitivity runs are easily set up but need to be designed carefully and the underlying assumptions to be properly justified. As mentioned above, this will be the subject of a companion paper to be submitted soon. The sensitivity experiments which we already performed (see reply to reviewer comment 1) clearly reveal that changing our assumptions on the soot impact on cirrus formation would have only marginal effects on the key model results and related conclusions presented in this paper.

3) In the light of my first major comment, I strongly suggest to use a model version without the assumption of soot being an efficient INP as the base version, and to present the simulation with soot INP as a sensitivity study.

As the goal of this study was to evaluate the new model version in the same configuration to be used for aviation studies, we believe that the current setup is appropriate. We further refer to our replies to the reviewer's comment 1).

4) Fig 2: the very pronounced north-south asymmetry of LWP is inconsistent with observations and with the bulk of CMIP5 models. This indicates that the cloud lifetime effect is probably too strong in EMAC-MADE3. Please discuss.

This is indeed already mentioned in the text (page 18 of the revised version): *"Most striking is the high bias in the Northwest Pacific ocean, which may be related to a high bias in the cloud lifetime in this region"*. Other possible sources of biases (CDNC, low cloud fraction) are however not to be excluded and are mentioned as well.

5) On the one hand, I appreciate that you chose a lower CDNCmin than ECHAM-HAM (Neubauer et al., 2019, doi:10.5194/gmd-12-3609-2019, see also https://doi.org/10.5194/gmd-2018-307-RC1 on this point). On the other hand, I wonder how you can justify a radiative imbalance as large as 5 W/m^2. Please explain.

Note that in the new setup with L41 vertical resolution and the revised tuning, the model imbalance is reduced to 3.4 W m$^{-2}$. As now discussed in Sect. 4.1, a lower imbalance could be obtained choosing a different set of parameters, but at the expenses of a worse agreement with the observation for other essential variables, such as liquid water path and cloud radiative forcing. Since we are not aiming at coupling this version with an ocean model, nor at running long-term climate simulations, but rather to perform short-term, nudged time-slice experiments on specific processes, our tuning strategies has been focusing on essential cloud and radiation variables, as also mentioned in the introduction (*"The new model configuration is tuned to obtain an optimal agreement in the representation of key cloud and radiation variables in comparison with observations"*). In this regard, we believe a 3.4 W m$^{-2}$ imbalance can be considered acceptable, also in view of the fact that the simulation discussed here was performed in nudged mode and that the corresponding free running experiment has an imbalance of only -0.5 W m$^{-2}$. This was also found by Schultz et al. (*Geosci. Model Dev.*, 2018) when comparing different ECHAM6 simulations with and without temperature nudging. We further elaborate on this issue in the revised version of Sect. 4.1: *"We recall that the EMAC-MADE3 configuration being tuned in this work is nudged, i.e. meteorological variables such as temperature, winds, and the logarithm of surface pressure are relaxed towards reanalysis data. In line with the designed application target of this version of the model, this allows to run different simulations (such as perturbation experiments) with very similar meteorological conditions. Such a model setup is most suitable for short-term time-slice experiments, that aim at isolating the effects of specific sources and processes, which would be statistically and numerically far more challenging without nudging. The use of the nudging technique has to be kept in mind while tuning the model, since nudging unavoidably impacts on the model climate, as it introduces a forcing component by modifying the model's temperature profile, which in turn perturbs the radiative balance (Zhang et al., 2014). A previous study by Schultz et al. (2018) based on the ECHAM6 model showed that temperature nudging may introduce a radiative imbalance around 5 W m$^{-2}$ with respect to an otherwise identical configuration without nudging. In the following, we will apply our tuning procedure to the nudged setup. Once the optimal configuration is identified, we will additionally perform a control experiment in free running mode to address the above issue and quantify the actual impact of nudging on the radiative balance in the tuned model configuration.*
*[…]*
*Deciding on a reference set of tuning parameters always involves expert judgment and a compromise between certain basic principles (Schmidt et al., 2017), as the choice of priorities must be guided by the main application target of the model setup to be tuned. If, for instance, the model is to be coupled to an interactive ocean, a close-to-zero radiation balance at the top of the atmosphere must be the central goal of tuning. In this study, the first priority has been laid on providing a reasonably good agreement with the observed values of the main cloud and radiation variables, although the model retains a radiative imbalance of 3.4 W m$^{-2}$. A lower imbalance could be obtained choosing a different set of tuning parameters, but at the expense of a worse agreement with the observations of other essential variables. This has been tested by performing two additional tuning simulations, increasing CDNCmin to 40 cm$^{-3}$ and reducing dnuc to 5 nm with respect to the reference set,*

*respectively. The results, shown by the blue markers in Fig. 1, demonstrate that this choice would improve the radiative balance to a value of 0.1 W m$^{-2}$ in both cases, but would also lead to larger errors in most of the other variables. As discussed above, however, a significant imbalance is introduced by temperature nudging, which is applied in all experiments shown here. To quantify this, a simulation with the tuned configuration but in free running mode is performed, which encouragingly results in a radiative balance of −0.5 W m$^{-2}$. Hence, while the nudged configuration – with its imbalance of 3.4 W m$^{-2}$ – is well suited for its designed purpose, also the free running model would meet the common requirements. We stress again that introducing a change of a few W m$^{-2}$ on the radiative balance through nudging is fully consistent with the study by Schultz et al. (2018) mentioned above.*

6) Fig 5: Observations suggest a clear dependence of R_ice on temperature (e.g. lower panel in Fig. 5c), which I have also seen reproduced in a global model with a two-moment ice microphysics (Salzmann et al., 2010, doi:10.5194/acp-10-8037-2010). EMAC-MADE3, on the other hand, does not reproduce this dependence. Instead, EMAC-MADE3 slightly overestimates R_ice at very low temperatures and strongly underestimates R_ice at higher temperatures. Both seems consistent with various heterogeneous nucleation processes being too efficient. (Overly efficient heterogeneous nucleation at very low temperatures could in principle suppress homogeneous nucleation, which can lead to an underestimate of the ice numbers.). Too small ice crystals in the mixed phase regime may also affect LW-ERF estimates. Please discuss.

As mentioned above, we now discuss a model simulation with increased vertical resolution which significantly reduces this bias. However, the problem is much more complex and this bias cannot be simply ascribed to a single process. In one of the many test simulations performed for this study, for example, we tried switching off heterogeneous nucleation in cirrus clouds, but this did not improve the comparison with observations. The model, however, shows a significantly better performance when increasing the vertical resolution. Therefore we think that the dynamical forcing and the representation of vertical velocities are more important aspects than INPs for an accurate representation of cirrus properties.

Perhaps move p. 3, l 26 ff to p3, l7? On the other hand, it would be better to re-write this entire paragraph taking into account the major comments above.

We rewrote this part as suggested in the major comment 1) (see reply above), taking into account this remark as well. Thank you for this suggestion.

p. 19, line 9: −1.76±0.04 Wm$^{-2}$: please discuss this result in the light of major points 2,3, and 4.

We added a sentence to relate the quite strong aerosol ERF to the bias in LWP (*"The high bias in liquid water path discussed in Sect. 4.2 and shown in Fig. 2 could be responsible for this high sensitivity, due to an overestimated cloud lifetime effect"*) and another sentence to stress once again that a detailed analysis of the impact of (aviation) soot on RF will be included in a follow-up study (*"The quantification of the cirrus effect under different assumptions on the ice nucleating properties of BC from various sources and other relevant ice nuclei is intended to be part of a follow up study."*). Note, however, that with the updated model configuration at higher vertical resolution, the resulting RF is lower (-1.42±0.02 W m$^{-2}$) and much closer to several other models estimates.

Fig 2: it would be good to also show zonal mean LWP plots.

The zonal mean plots would contain less information than the current lat-lon plots. Furthermore, averaging along the longitude coordinate could hide some of the biases due to error cancellation. We would therefore refrain from including such plots.

p. 1, lines 9 to 11 ("The performance ..."): please refer to my major comment #4 regarding the hemispheric asymmetry of the simulated LWP above.

We now explicitly mention the LWP bias in the abstract: *"Some remaining discrepancies, namely a high positive bias in liquid water path in the northern hemisphere and overestimated (underestimated) cloud droplet number*

*concentrations over the tropical oceans (in the extra-tropical regions), which are both a common problem of this kind of models, need to be taken into account in future applications of the model"*

p. 1, line 10f: please be specific

We would refrain from adding too many details in the abstract.

p. 1, lines 12ff: please refer to my major comments #4 and #5 above and to my comment regarding p. 20, line 13.

The following sentence has been added at the end of the abstract, trying to summarize the above replies to these comments: *"…an estimate of the anthropogenic aerosol effective radiative forcing (ERF) is provided, showing that EMAC-MADE3 simulates a relatively strong aerosol induced cooling, but within the range reported in the IPCC assessments."*

p. 7, lines 14ff: "Since the EMAC-MADE3 ….": this seems pure speculation. A least one would have to show that these biases do not occur in models that use different emission data sets in a comparable AMIP-style setup.

We agree and removed the sentence, although we kept the part about emission uncertainties as a possible reason for these biases. This has now been moved to the previous sentence: *"Uncertainties in the prescribed emission fluxes could also contribute to these biases, especially in East Asia, where anthropogenic emissions in the year 2000 are higher than in other regions of the world and have further increased since then."*

Fig 3: since the model values are mean values, I suggest to include vertical bars which give some measure of the spread between the model values (e.g. standard deviations or quantiles).

Thank you for this suggestion: we added vertical bars to Fig. 3 to show the standard deviation calculated from the model interannual variability.

p. 10, l. 6: as an aside: rather than only matching observations it would perhaps be better to include a process-oriented approach in the future. For example, if the warm rain formation rate is overestimated compared to observations and too little rain forms via the ice phase, this indicates that the cloud lifetime effect may be overestimated. On the other hand, tuning strategies taking such process-oriented metrics into account are only starting to emerge.

Thank you for this suggestion. We will consider this possibility in future evaluation studies of EMAC-MADE3.

p. 17, l. 2: would it be feasible to regrid the observations to model resolution in order to avoid this problem?

The observational data are not spatially gridded, but provided as probability distribution functions in 1-K temperature bins. Therefore the variability is intrinsic to the distribution and cannot be removed. It is also a valuable piece of information that should not get lost by applying, for instance, a vertical binning of the data within the model's vertical levels. Another complication arises from the fact that cirrus clouds are characterized by a quite sparse distribution and a regridding/binning procedure might introduce artifacts and lead to misinterpretation of the data. By plotting the data as we did in Fig. 5, we show the actual distributions of both model and observation data, minimizing the loss of information due to data processing.

p. 17, l. 12: the simulated RH is also controlled by the large-scale condensation scheme, the assumption regarding cloud phase in the saturation adjustment, and other microphysical terms. In the original ECHAM5, the assumption regarding cloud phase in the saturation adjustment is based on a threshold for the mass mixing ratio of cloud ice, but this is different in EMAC. Line 18 on page 15 says that EMAC allows supersaturation over ice based on Murphy and Koop (2005) and supersaturation over ice is also seen in Fig. 6c. The simulated supersaturation depends on this parameterization.

Thank you for pointing this out: we added the citation to Murphy and Koop in this sentence to stress this again.

p. 18: line 15: are you comparing a model with prescribed SSTs to the CMIP5 coupled models or is precipitation reproduced remarkably well also when compared to AMIP style simulations with other models?

We are not sure we understand this comment: the comparison of precipitation is done with respect to GPCP-SG data in Sect. 4.5 and we do not mention the CMIP5 models in this context.

We removed the last sentence of this paragraph.

p. 19, l. 6: I tend to think of the aerosol radiative forcing as the direct forcing. I think what you computed is better called effective radiative forcing (ERF), which by definition includes fast adjustments.
Thank you for noting this inconsistency. We corrected RF to ERF in the text.

p. 19, l. 5: I assume you are using the standard double calls to estimate cloud radiative effects. Or does Dietmüller et al. (2016) do something special? If yes, please explain. If no, please mention that this is a standard method, so that people who know this method won't have to check Dietmüller et al. for details.
Yes, it is a standard double call method. The Dietmüller et al. (2016) implements it in an extended way, allowing multiple calls with different forcing agents. However, this is not relevant here and to avoid confusion we have removed the citation to Dietmüller et al., which is anyway already provided in the model description section.

p. 20, line 13: "the total aerosol RF reported by the IPCC ranges between −2.05 and 0.05 Wm−2". Please be specific. How does this fit with AR5 Table 7.4? Also note that in the model with the largest ERFari+aci in AR5 Table 7.4, the ocean heats the atmosphere in the 20th century (Fig. 4, Golaz et al., 2013, doi.org/10.1002/grl.50232) in a setup that uses basically the same aerosol activation scheme, but a different microphysics scheme.
The original values were taken from Table 8.6, i.e. from Chapter 8, as also referenced in the text (Myhre et al., 2013). However, in the revised version of the paper we replaced these numbers with the assessment based on expert judgement in IPCC Chapter 7, as suggested: *"A comparison with the estimates of the IPCC AR5 (Boucher et al., 2013; Myhre et al., 2013) shows that EMAC-MADE3 simulates a more negative aerosol effect: the aerosol ERF (sum of the effects from aerosol-radiation and aerosol-cloud interactions) reported by the IPCC and based on expert judgement is −0.9 W m$^{-2}$ with a 5 to 95% uncertainty range of −1.9 to −0.1 W m$^{-2}$."*
We also added the estimate from a recently published review by Bellouin et al. (*Rev. Geophys.*, 2019), which considers observational constraints and basically confirms the CMIP5 range: *"The recent review by Bellouin et al. (2019) considers additional observational constrains on the aerosol ERF and reports a range of −1.60 to −0.65 W m$^{-2}$ (−2.0 to −0.4 W m$^{-2}$) for a 68% (90%) confidence interval, respectively, which basically confirms the IPCC ranges."*

p. 21, l. 3: tuned for what? Usually, I would think about TOA balance. But the model is very much out of balance. I think you need to make clear that by tuned you mean tuned mainly to match LWP, IWP, ICNC, SWCRE, and LWCRE to values that were derived from satellite (remote sensing) observations. The large TOA imbalance of 5.23 W/m^2 precludes a successful coupling to an ocean component without further (fairly drastic) tuning measures.
As discussed in the comment above, our tuning strategies targeted main cloud and radiation variables, while accepting a (now smaller) radiative imbalance of 3.4 W/m$^2$ since no coupling of this model configuration to an ocean component is planned. This is now explained in more detail in Sect. 4.1.

p. 21, l. 7: satellite measurements: space-borne instruments measure radiation. Not even clear-sky radiation in a cloudy scene (as used to compute CREs in the model) is among the measured quantities.
Good point. We replaced *"satellite measurements"* with *"satellite data"*.

p. 21, list item 1: please see my major comment #4 above.
The major biases are discussed in item 2 of the list which has been rephrased in view of comment 4) as follows:
*"specific deviations, in particular in the representation of liquid water path which could point to an*

*overestimated cloud lifetime, mostly confirm known biases of the ECHAM5 model and can therefore not be attributed to the new cloud scheme introduced in this work;"*

p. 21, list item 3: please see my major comment #6 above. (I realize that a part of this point is implicitly included in the last sentence. But I still think the unrealistically weak dependence of effective radius on temperature is notable. )
As discussed, we significantly reduced this bias by increasing the model vertical resolution.

p. 21, list item 4: here you are explicitly referring to "coupled" models from CMIP5. In the context of CMIP5, "coupled" usually refers to the fact that the models are run with an ocean component. Because EMAC-MADE3 is run in atmosphere-only (AMIP) mode (with fixed SSTs), cloud biases (e.g. the double ITCZ) should definitely be smaller than in coupled models. Note, however that in addition to results from coupled model experiments, the CMIP5 archive also includes results from dedicated atmosphere-only (AMIP) model experiments. In order to justify the statement that you are making here, you should compare EMAC-MADE either to uncoupled CMIP5 models (AMIP runs) or else to uncoupled (i.e. without ocean component) AeroCom experiments.
Thank you for this remark. Indeed the term "coupled model" is ambiguous in this context. We have therefore rephrased this sentence as follows: *"the overall performance of EMAC-MADE3 in simulating global cloud and radiation variables is in line with the results of the CMIP5 models"*. As reported by Lauer et al. (*J. Clim.*, 2013), *"the AMIP models do not systematically outperform the coupled models in reproducing observed mean cloud properties. Furthermore, this suggests that the large intermodal spread in total cloud amount and liquid water path is attributable to the representation of cloud processes rather than to biases in the SST and related aspects of the circulation in the coupled models"*. This is also shown in Fig. 5 and Fig. 6 of the same paper, which supports our statement in the conclusions.

p. 21, list item 5: please be specific
We now explicitly mention the 3 models which are discussed at the end of Sect. 4.4, namely ECHAM5-HAM, EMAC-GMXe, and NCAR-CAM3.5.

p. 21, item 6: where is this shown?
This analysis was performed offline, but since this point is not relevant here we removed it from the list. Details of the aerosol evaluation can be found in Kaiser et al. (*Geosci. Model Dev.*, 2019). This study is referenced in several parts of the present study (as K19).

p. 21, line 4 from bottom: please see my comment regarding p. 20, line 13.
As mentioned above, the IPCC results which we are referring to are taken from Table 8.4 in Chapter 8 (Myhre et al., 2013) and are further supported by Bellouin et al. (2019).

p. 21, line 2 from bottom: please see my major comment #1.
As discussed above and also mentioned here, additional simulations targeting the BC-cirrus effects will be presented in a follow-up study.

p 23ff: it would be nice to analyze how much each of the processes contributes to the ice crystal number at a given temperature. One could for example plot time averaged process rates on the x-axis and temperature on the y-axis.
This analysis is certainly interesting but is not within the scope of this paper, also because the role of the different processes in the cirrus parametrization was extensively analysed in Kuebbeler et al. (*Atmos. Chem. Phys.*, 2014), using the ECHAM5-HAM model. Their study is referenced in our paper.

The conclusion section should include a discussion of uncertainties which are specific to this new model development effort. Furthermore, there are some general caveats to this type of study that could be mentioned in the introduction section. In particular, numerous studies suggest that in cases in which the cloud

lifetime is limited by turbulent mixing and not by precipitation, coarse scale models may overestimate the cloud lifetime effect (see e.g. references in Salzmann et al., 2010, doi:10.5194/acp-10-8037-2010). Other studies have shown that the aci component of ERFari+aci in one model depends on the CDNCmin tuning parameter (Neubauer et al., 2019, doi:10.5194/gmd- 12-3609-2019, Hoose et al., 2009, doi:10.1029/2009gl038568) and in another model on the autoconversion threshold (Golaz et al., 2010, doi: 10.1175/2010JCLI3945.1).

Thank you for suggesting these references. We added them in Sect. 5 in the context of the ERF discussion: *"This result shows that EMAC-MADE3 tends to simulate a comparatively high aerosol-induced cooling and so it could be more sensitive to changes in aerosol concentrations than other global aerosol climate models. The high bias in liquid water path discussed in Sect. 4.2 and shown in Fig. 2 could be responsible for this high sensitivity, due to an overestimated cloud lifetime effect. This is a common problem in coarse-resolution global models, which are not able to resolve the enhanced entrainment of dry air in clouds with higher CDNC, which would lead to droplet evaporation and thus partly offset the CDNC-induced increase in cloud lifetime (Ackerman et al., 2004; Xue and Feingold, 2006; Jiang et al., 2006; Altaratz et al., 2008; Mülmenstädt et al., 2019). Other studies show that the choice of the tuning parameters can also determine the resulting aerosol ERF: Hoose et al. (2009) and Neubauer et al. (2019) found a smaller (i.e., less negative) aerosol ERF when increasing $CDNC_{min}$ from 10 $cm^{-3}$ (–1.7 W $m^{-2}$) to 40 $cm^{-3}$ (–1.0 W $m^{-2}$) in the ECHAM6 model, while Golaz et al. (2011) reported a linear dependency of the aerosol radiative flux perturbation on the autoconversion threshold radius in the GFDL AM3 model.".*

As suggested, we added a concluding paragraph to summarize the main uncertainties specific to this study: *"Despite the encouraging performance of this new model configuration, several uncertainties remain and have to be addressed in the future by targeted applications of EMAC-MADE3. This includes, for instance, the tendency of the model to simulate a relatively large anthropogenic aerosol radiative forcing, which is a common feature in coarse-resolution global models and may affect the estimates of the climate impacts of specific sectors, such as aviation. The model biases in CDNC and ICNC could also influence the model's ability to reproduce observed cloud radiative properties. Another limitation is related to the uncertain properties of ice nucleating particles, which in the case of black carbon still lack a solid experimental confirmation. Furthermore, the cirrus parametrization implemented here is based on a supersaturation threshold for ice nucleation, but it is not able to follow the ice formation process in detail by means of, e.g., a nucleation spectrum. This means that a single critical value is provided for each ice nucleating particle type, but this does not fully represent the complexity of the actual physical process. Finally, in this study we focused on ice formation in cirrus clouds from the perspective of aerosol particles driving this process, but it should not be forgotten that mesoscale and large-scale atmospheric dynamics plays an equally (or even more) important role in the microphysics of cirrus clouds."*

Technical: p. 8, l. 19: calculate -> calculated

Fixed. Thank you for spotting.

Correct line numbers would have helped with the review.

We agree. Apparently something went wrong during the production stage, since the line numbers were displayed correctly in the submitted PDF version.

**Anonymous Referee #2**

1. The paper is all about clouds and aerosols, but I cannot find how activated aerosols are connected to cloud and crystal number. Please state that explicitly.

The calculation of cloud droplet number is described in Sect. 3: *"Formation of cloud droplets is described following Abdul-Razzak and Ghan (2000), calculating the fraction of activated aerosol particles at a given supersaturation as a function of their radius and composition. Here, it is assumed that only the soluble compounds ($SO_4^{2-}$, $NO_3^-$, $NH4^+$, $Na^+$ and $Cl^-$) contribute to the mean hygroscopicity parameter which controls the critical supersaturation for particle activation. An alternative formulation by Petters and Kreidenweis (2007) to calculate the supersaturation based on a single $\kappa$ parameter has also been implemented in EMAC and coupled to MADE3 in this work. Results of sensitivity simulations (see Sect. 4.3), however, revealed no significant differences in cloud droplet number concentration (CDNC) obtained with the different approaches and the Abdul-Razzak and Ghan (2000) approach is used to calculate the supersaturation in the simulation evaluated in this work."*

The calculation of ice crystal number was originally described in Appendix A, but has now been extended and moved to Sect. 3 to address Comment 2 below.

2. The information in Appendix A is only referred to in passing, and should be a larger part of the manuscript.

Thank you for this suggestion. We have moved Appendix A to the main text as a new subsection (Sect. 3.1) and we also extended the description of the cirrus parametrization in Sect. 3 as follows: *"The cirrus parametrization by Kärcher et al. (2006) implements the competition among the different ice formation processes in decreasing order of efficiency, i.e. from the heterogeneous freezing of the INP with the lowest $S_c$ to homogeneous freezing. Due to a cooling of air parcels induced by updrafts, $S_c$ increases until the consumption of supersaturated water vapour by the growth of freshly formed or pre-existing ice crystals is large enough to terminate this process. The consumption of water vapour via depositional growth of pre-existing ice is accounted for by reducing the vertical velocity in Eq. (1) by a so-called fictitious downdraft. If the cooling rate which corresponds to the reduced vertical velocity is still large enough to generate sufficiently high supersaturations, the heterogeneous and homogeneous ice formation processes, and the competition among them, can take place. Ice crystals larger than 100 μm, typically formed by aggregation, are transferred to snow crystals which are assumed to be removed within one model time-step by precipitation, melting or sublimation (Levkov et al., 1992). This is introduced to avoid model instabilities which might arise due to a too fast sedimentation of large ice crystals (K14). Multiple ice modes (heterogeneous and homogeneous) are considered only for the ice nucleation and depositional growth processes, while for aggregation, accretion and transport a unimodal approach is used. The resulting ice crystal number concentration and ice water content is given by the sum of the concentrations in the individual ice modes. Further details on the cirrus parametrization, including the results of a box-model simulation, can be found in Kuebbeler (2013) and K14."*

3. The sensitivity tests and tuning exercises are laudable, but the description of optimizing the model and what parameters are chosen is ad-hoc and subjective. It doesn't seem like any formulation matched the data, but no rigorous method was used to get the final tuned model.

As mentioned at the beginning of Sect. 4.1, the tuning method is based on Lohmann and Ferrachat (2010) and analogous approaches have also been used in recent evaluation studies of global models with aerosol-cloud couplings (e.g., Neubauer et al., *Geosci. Model Dev.*, 2019). In the revised version of the manuscript, we improved this approach by introducing quantitative statistical measures (the normalized RMSE and the relative standard deviation) to better motivate our choice of the parameters. Note, however, that model tuning is always subjective to some extent, as it depends on the application target of the model under investigation. In the revised Sect. 4.1, we added a paragraph to clarify this in more detail: *"Deciding on a reference set of tuning parameters always involves expert judgment and a compromise between certain basic principles (Schmidt et al., 2017), as the choice of priorities must be guided by the main application target of the model setup to be tuned. If, for instance, the model is to be coupled to an interactive ocean, a close-to-zero radiation balance at the top of the atmosphere must be the central goal of tuning. In this study, the first priority has been laid on providing*

*a reasonably good agreement with the observed values of the main cloud and radiation variables, although the model retains a radiative imbalance of 3.4 W m$^{-2}$. A lower imbalance could be obtained choosing a different set of tuning parameters, but at the expense of a worse agreement with the observations of other essential variables. This has been tested by performing two additional tuning simulations, increasing CDNCmin to 40 cm$^{-3}$ and reducing $d_{nuc}$ to 5 nm with respect to the reference set, respectively. The results, shown by the blue markers in Fig. 1, demonstrate that this choice would improve the radiative balance to a value of 0.1 W m$^{-2}$ in both cases, but would also lead to larger errors in most of the other variables. As discussed above, however, a significant imbalance is introduced by temperature nudging, which is applied in all experiments discussed here. To quantify this, a simulation with the tuned configuration but in free running mode is performed, which encouragingly results in a radiative balance of −0.5 W m$^{-2}$. Hence, while the nudged configuration − with its imbalance of 3.4 W m$^{-2}$ − is well suited for its designed purpose, also the free running model would meet the common requirements. We stress again that introducing a change of a few W m$^{-2}$ on the radiative balance through nudging is fully consistent with the study by Schultz et al. (2018) mentioned above."*

4. Some of the plots show significant artifacts (oscillations) in averaged model fields. I am hoping that this is just a product of the 11 hour time sampling, but it means there is either something fundamentally wrong with either the model itself or the analysis methods.

We are not sure to which plots the reviewer refers to here. Given a similar comment below ("Page 11, L4"), we argue this is about Fig. 2, 7 and 8. These wave patterns are typical of spectral models such as ECHAM5, which is the base model of EMAC. It has nothing to do with time sampling, as these patterns are visible also in the raw model output at the model time-step level. See an example below for surface pressure:

[Figure]

5. Since this is about ice clouds and aerosols, please determine the ice cloud ERF from anthropogenic aerosols. Techniques exist for this.

We are aware of such techniques, but as discussed above in the replies to Reviewer #1 the goal of this publication is to document and evaluate a new model configuration to be applied in future studies on the impact of anthropogenic soot and aviation soot on cirrus clouds. Additional sensitivity runs are easily set up but need to be designed carefully and the underlying assumptions to be properly justified. As mentioned above, this will be the subject of a companion paper to be submitted soon.

6. Substantive comments below on particular aspects of the data analysis also need to be considered in a revision.

See detailed replies below.

Page 3, L16: this does not go back far enough. Liu et al (2005) developed a parameterization of homogeneous and heterogeneous freezing that was implemented in CAM by Gettelman et al (2010).

Thank you for suggesting this reference. Also in view of the comments by Reviewer #1, we have extended this paragraph to include more studies. Both Liu et al. (2005) and Gettelman et al. (2010) are now cited.

Page 3, L34: The abstract mentions the mixed phase, but there is no discussion here. This seems to be all cirrus clouds.

Yes, indeed. The focus of the paper is on a new model configuration targeting aerosol effects on cirrus, but as also explained in Sect. 3, the model includes a parametrization for mixed-phase and liquid clouds as well.

Page 6, L33: please state how the aerosols and clouds are connected. You detail how INP are derived. But once you have INP how are they linked with the cloud scheme? Do you just set the cloud drop and crystal number to be at least as large as the activated number?
Thank you for this comment. This was admittedly not detailed enough and has been extended (see reply to major comment #2 for details).

Page 7, L14: since the INP parameters are the new part of this study, I think a bit more detail is warranted here, maybe a paragraph, and not left for the appendix.
As mentioned in the reply to major comment #2, Appendix A is now part of the main text (Sect. 3.1).

Page 8, L4: so what are you really focusing on? Not clouds? Not aerosols? What do you mean by the 'interaction'. At this point as a reader I still have not idea. Please be more specific.
We agree with the reviewer that this part was confusing and have rephrased the paragraph as follows: *"The conclusions of the K19 evaluation on the aerosol representation in the uncoupled model version still hold for the aerosol-climate coupled version discussed here, since the aerosol-cloud and aerosol-radiation couplings do not lead to significant changes in the global aerosol characteristics. However, the present configuration uses a higher vertical resolution than in K19, with 41 instead of 19 vertical levels. This leads to some differences in the representation of aerosol in the cirrus-relevant upper tropospheric regions, but showing in most cases a slightly improved model performance in these regions (see Figs. S1 and S2 in the Supplement). In this study, we focus on cloud and radiation variables and analyze the performance of EMAC-MADE3 in reproducing essential quantities (such as total cloud cover, cloud liquid and ice water, cloud droplet and ice crystal number concentrations, and cloud radiative effects) compared to satellite and in situ observations. Since the present configuration is developed with a specific focus on cirrus clouds, a special attention will be devoted to this aspect."*

Page 8, L23: please at least mention what the data sources are (CERES, MODIS, etc)
W added a table (Table 2) to summarize the datasets used for tuning, with the corresponding references.

Page 8, L24: so how many simulations were conducted? 20? How long are they? Why not use a more sophisticated method for sampling that can better sample a wider parameter space following Carslaw, Regare, etc.?
We added a sentence in the text to better clarify the number of experiments required for tuning: *"We tested 5 values for each of the 4 tuning parameters $\gamma_r$, $\gamma_s$, $CDNC_{min}$ and $d_{nuc}$, varying across a range of approximately one order of magnitude. We then calculated their effect on six cloud variables (total cloud cover, liquid water path (LWP) over the oceans, CDNC over the oceans, ice water content in cirrus clouds ($IWC_{cirrus}$), ice crystal number concentration in cirrus clouds ($ICNC_{cirrus}$), and precipitation) and three radiation variables (shortwave and longwave cloud radiative effects (SWCRE and LWCRE, as the difference between all-sky and clear-sky radiation fields), as well as the net radiative balance). Note that exploring the full parameter space, i.e. all possible combinations of the 5 values for the 4 tuning parameters, is not feasible as it would require performing $5^4 = 625$ model simulations. So we only explore the model sensitivity for each single parameter while keeping the others fixed at a reference value, which corresponds to the median of the explored range, i.e. $\gamma_r = 8$, $\gamma_s = 800$, $CDNC_{min} = 10\ cm^{-3}$, and $d_{nuc} = 10$ nm. This limits the numbers of simulations to be performed to 17 (i.e., $5 \times 4 = 20$ simulations minus 3 redundant cases). To further reduce the computational costs, the tuning simulations cover a period of only three years (1999-2001)."*

Page 9, Fig1: LWP needs a range of uncertainty.
Fig. 1 has been revised and now shows the normalized RMSE instead of the mean. The mean values of the evaluated variables are given in Table 3 (as in the previous version of the manuscript) and the corresponding observational uncertainty ranges are now provided for all of them, including LWP.

Page 10, L5: so was there an objective way the parameters are chosen? It's not clear how you made a decision. Wouldn't a root mean square error be better than means? How do you know there is not cancellation?

Motivated by this comment we completely revised the tuning approach and we now use the normalized RMSE and the relative standard deviation as more objective metrics to identify the optimal setup. We also rewrote Sect. 4.1 to better motivate our choice of the tuning parameters and we ran additional sensitivity simulations to support our decision when the tuning experiments do not give a clear indication and trade-offs between the variables are present.

Page 11, L4: Cloud fraction should only be compared if the model and the data are thresholded identically, usually requiring use of a satellite simulator. Why is there a significant oscillation in mean cloud fraction and LWP fields over the subtropical oceans in the model? That needs to be explained.

A satellite simulator is unfortunately not available for the model configuration presented in this study, but will be considered in future evaluations of EMAC-MADE3. Note, however, that Lauer et al. (*J. Clim.*, 2013) found that the performance of most CMIP5 models in reproducing total cloud cover does not improve considerably when using the COSP simulator. Regarding the oscillation in the cloud cover and LWP fields, we refer to the reply to the major comment 4 above.

Page 13, Figure 3: gray does not stand out from dark blue. I can not distinguish the DACCIWA point. Also, can you give some idea of regime or location, maybe another plot with difference from the 1:1 line as a function of latitude?

Thank you for noting this: we have changed the dark blue in Fig. 3 to a lighter blue that should be better distinguishable from gray. The location/regime can already be differentiated by the three groups of measurements color coded: clean marine, polluted marine, continental. We chose this method to allow for a direct comparison with other studies using the same data, e.g. Karydis et al. (2017) and Rothenberg et al. (2018).

Page 14, L5: so the reason for the high SWCRE is likely the high CDNC? Is that a correct statement?

This could be a possibility, but we would refrain from including such a statement in the paper.

Page 14, L14: in figure 4, the gradients are wrong in EMAC over most of the oceans. CDNC drop with latitude in EMAC but increase with latitude in the observations. What is going on? This seems like a fundamental error somewhere either in model or data that needs explanation.

We looked deeper into the CDNC evaluation and extended it by comparing with a further satellite dataset (Grosvenor and Wood, 2014, 2018), see Fig. 4. We also extended Sect. 4.3 with a discussion on the observational uncertainty and related our model's performance to other models reported in the literature. The corresponding text in Sect. 4.3 now reads as follows: *"We also compare the simulated CDNC with satellite retrievals, which provide a unique global picture of this quantity, although this kind of retrievals are still affected by considerable uncertainties (Grosvenor et al., 2018). Here we use a recent 13-year climatology by Bennartz and Rausch (2017), based on MODIS-AQUA retrievals. In-cloud CDNC, as reported in the observational dataset, are selected with the same method used for comparing with in situ data, but considering CDNC at cloud top, as observed by the satellite, i.e. by taking the CDNC in the highest model level with a liquid cloud. An alternative method, taking the average CDNC through the cloudy part of the column, provides very similar results (not shown). This is expected, since the representation of liquid cloud formation in the EMAC cloud scheme follows the adiabatic parcel theory, assuming that newly formed cloud droplets at the cloud base are equally distributed in the vertical by mixing, regardless of the aerosol concentrations. An identical assumption is also done in the retrieval process by Bennartz and Rausch (2017). The results of this comparison are depicted in Fig. 4. In the Northern Hemisphere, EMAC captures the major spots of high CDNC over the Atlantic Ocean eastward of Canada and USA, over the Mediterranean and eastward of China, albeit with about 50% higher CDNC than MODIS. These spots also have a wider horizontal extent over the oceans than in the observations. This could be due to the generally high CDNC over the continents (as shown by the in situ data in Fig. 3) being too efficiently advected over the oceans or, as mentioned in Sect. 4.2, to a bias in the prescribed emissions,*

[revised manuscript text omitted]

Thanks for the suggestion about the variability: we have added the lines for the 25/75% quantiles of the model in addition to the median in Fig. 5 and Fig. 6 (red dashed lines)

Page 18, L15: for Figure 7 and all the subsequent maps: can you calculate root mean square error for all the fields. This is useful for quantitatively understanding pattern biases.

The normalized mean bias for the reference and the tuning simulation is now shown in Fig. 1 for all evaluated variables, in the context of the revised tuning approach discussed above.

Page 19, Fig 7: why is EMAC so noisy in its cloud fields? If it is sampling then you should use a different method. If it is not a sampling issue, then something is wrong with the model. I've seen this before with radiation code errors leading to jumps in solar zenith angle and incident radiation...

See reply to major comment 4 above. This is a feature of the ECHAM5 spectral core.

Page 19, L11: does this mean the cloud effect is -1.13 and the rest (-0.63) is a direct effect? That seems large.

Note that in the revised configuration the cloud effect is now -0.96 W m$^{-2}$. It is indeed quite large, as we explicitly admit in the same paragraph, which has now been extended to discuss possible reasons behind this relatively high value: *"This result shows that EMAC-MADE3 tends to simulate a comparatively high aerosol-induced cooling and so it could be more sensitive to changes in aerosol concentrations than other global aerosol climate models. The high bias in liquid water path discussed in Sect. 4.2 and shown in Fig. 2 could be responsible for this high sensitivity, due to an overestimated cloud lifetime effect. This is a common problem in coarse-resolution global models, which are not able to resolve the enhanced entrainment of dry air in clouds with higher CDNC, which would lead to droplet evaporation and thus partly offset the CDNC-induced increase cloud lifetime (Ackerman et al., 2004; Xue and Feingold, 2006; Jiang et al., 2006; Altaratz et al., 2008; Mülmenstädt et al., 2019). Other studies show that the choice of the tuning parameters can also determine the resulting aerosol ERF: Hoose et al. (2009) and Neubauer et al. (2019) found a smaller (i.e., less negative) aerosol ERF when increasing CDNCmin from 10 cm$^{-3}$ (−1.7 W m$^{-2}$) to 40 cm$^{-3}$ (−1.0 W m$^{-2}$) in the ECHAM6 model, while Golaz et al. (2011) reported a linear dependency of the aerosol radiative flux perturbation on the autoconversion threshold radius in the GFDL AM3 model."*

Also note that the clear-sky effect includes the direct aerosol effect as well as the radiative forcing resulting from the effect of the cloud modifications on water vapour. The individual contributions cannot be separated.

Page 20, L23: however, Gettelman et al 2012 found that the ice cloud indirect effect in CAM and ECHAM-HAM is positive. Please document the ice cloud portion of the ACI if you can (a simple simulation where you fix the IN and one where it varies) and compare to previous work, since that seems to be the major development in this paper. Gettelman, A., X. Liu, D. Barahona, U. Lohmann, and C. C. Chen. "Climate Impacts of Ice Nucleation." Journal of Geophysical Research 117, no. D20201 (2012). https://doi.org/10.1029/2012JD017950.

We would refrain from estimating the ACI portion due to cirrus clouds by means of a single sensitivity simulation. As stated in the paper and in the replies to the other reviewers, this is a very uncertain effect, which requires the exploration of the BC ice nucleating properties in details, by means of several, carefully designed and well-motivated sensitivity experiments. This will be part of a follow-up, dedicated application study.

Page 21, L31: I'm not sure how the parameters were chosen. This should be more objective

As stated above, we have completely revised our tuning approach, introducing quantitative metrics to objectively evaluate the tuning experiments and performed additional sensitivity simulations to explore the interdependencies of some tuning parameters in more detail. This is described in a revised and improved way in Sect. 4.1.

**Anonymous Referee #3**

1. As the authors state, the ice nucleating properties of BC are highly debated. In light of these uncertainties, it makes more sense to define the reference case without BC as INP and show a sensitivity case where the impact of BC as INP is included. How important is BC as INP using the current assumptions?

As discussed in the replies to reviewer #1 and #2, in this paper we evaluate the reference EMAC-MADE3 configuration which is going to serve as a basis for currently planned application studies. The inclusion of BC as INP is an essential part of this new configuration and is therefore documented here. Nevertheless, the analysis of its impact on, e.g., ICNC and climate, needs to be performed carefully, also in view of the large uncertainties on its ice nucleation properties. Performing a single sensitivity experiment is not sufficient and might produce misleading results. We believe, on the contrary, that a range of experiments exploring the many uncertain aspects of the BC ice nucleating properties is required and needs to be presented in one (or more) dedicated publication(s). This kind of analysis would also not fit to the scope of GMD.

2. The aerosol representation of MADE3 is quite complicated and an interesting feature of the work. I suggest explaining this in more detail (even though it has been explained in previous papers). In this context, the sentences on p. 23/lines 20-23 are hard to understand. Please clarify.

Thank for pointing us to this unclear part in the text. We have added a paragraph to introduce some information on the MADE3 model structure which should help the reader understanding the rest of this section (this now Sect. 3.1): *"As mentioned above, aerosol particles in MADE3 are distributed across three log-normal size modes (Aitken, accumulation and coarse mode), with three possible mixing states (soluble, insoluble, and mixed). Following the same notation as in K19, we indicate the MADE3 modes Aitken, accumulation and coarse with the index k, a and c, respectively. Mixing states are indicated by s, i, and m for soluble, insoluble and mixed, respectively. The relevant INPs considered in this study, namely BC and mineral dust (DU), are only present in the insoluble and mixed modes of MADE3 and mineral dust is only tracked in the accumulation and coarse mode. Of the 9 modes normally required in MADE3 for each aerosol compound, only 6 are hence needed for BC ($BC_{km}$, $BC_{ki}$, $BC_{am}$, $BC_{ai}$, $BC_{cm}$ and $BC_{ci}$) and 4 for mineral dust ($DU_{am}$, $DU_{ai}$, $DU_{cm}$ and $DU_{ci}$)."*

It would also be interesting to learn more about if keeping track of the aerosol mixing state as done with MADE3 (as opposed to only tracking average composition using fewer modes) is yielding improved results.

How the results depend on the mixing state resolution is indeed an interesting question. As mentioned in the introduction, the EMAC model includes another aerosol submodel (GMXe), which has a different representation of aerosol mixing states than MADE3 and more similar to HAM. Since this submodel has also been recently coupled to a cirrus parameterization (Bacer et al., *Geosci. Model Dev.*, 2019), a consistent comparison between the two approaches in the framework of the EMAC model infrastructure will be an interesting topic for a future evaluation study.

3. Given that this is a model development paper on a new cloud microphysical scheme, more emphasis should be given presenting the underlying set of equations. Some of this is currently presented in Appendix A. This should move to the main manuscript, but more needs to be added to explain how the "potential ice-nucleating particles" are used to produce ice crystals. Overall the presentation of the modeling equations is fragmented and hard to follow.

We agree with the reviewer that the model description section needed improvement. Considering also the remarks by reviewer #2, we have now moved Appendix A to the main text (Sect. 3.1) and extended the description of the cirrus parametrization in Sect. 3 to include more details, as suggested.

4. More details need to be provided for the model tuning in section 4.1. What is the justification to choose specifically these four tuning parameters? And given that for each of the four tuning parameters five values are chosen, do you perform all 20 simulations (one for each parameter calculation)? Have you considered using a latin hypercube sampling strategy of the parameter space of your tuning parameters? And on what basis is the optimal tuning configuration chosen?

For the tuning exercise we performed 17 simulations: 5 simulations times 4 parameters minus 3 redundant cases (since the central value of each parameter range is taken as reference and kept constant while varying the others). Two additional sensitivities have been run to investigate additional combinations of the parameters. This is now more clearly explained in the revised Sect. 4.1, where we also justify our choice of the parameters based on more quantitative metrics (NRMSE and relative standard deviation).

We are grateful to the reviewer for suggesting more advanced sampling methods. We will consider them in future tuning exercise with EMAC-MADE3.

5. How has including the new scheme changed the results compared to the previous version of the model? I.e. is the "new approach" an improvement over the old approach?

A systematic comparison with the previous version of the model (Lauer et al., *Atmos. Chem. Phys.*, 2007; Righi et al., *Environ. Sci. Technol.*, 2011; and Righi et al., *Atmos. Chem. Phys.*, 2013) would be challenging, since the new development presented here includes many other changes than just the new cloud scheme: we moved from the MESSy1 to MESSy2.5 environment and from MADE to MADE3 (as documented in Kaiser et al., 2019) and we now implemented a new cloud scheme coupling it to MADE3. There is also no previous version of this coupled configuration with MADE or MADE3 in the current MESSy environment.

The new approach is an improvement since it allows describing new processes (e.g., aerosol-induced ice formation in cirrus clouds) and therefore to perform studies which were not possible with the old approach.

So our point is not about comparing the performance of a new versus an old model version, but to include more processes showing that the model still delivers reliable results. Also note that with EMAC-MADE3, a more comprehensive evaluation of the cloud and radiation variables has been performed with respect to our previous studies, by including more and up-to-date datasets, considering more variables, and refining the tuning and evaluation techniques.